

# Microscopic polyangiitis plasma-derived exosomal miR-1287-5p induces endothelial inflammatory injury and neutrophil adhesion by targeting CBL

Yan Zhu[1,2], Liu Liu[1], Liepeng Chu[1], Jingjing Lan[1], Jingsi Wei[1], Wei Li[1] and Chao Xue[1]

[1] Department of Nephrology, The Second Affiliated Hospital of Guangxi Medical University, Nanning, Guangxi, China
[2] The First Affiliated Hospital, Department of Nephrology, Hengyang Medical School, University of South China, Hengyang, Hunan, China

## ABSTRACT

**Background**. An inflammatory environment around the vessel wall caused by leukocyte infiltration is one of the characteristic histopathological features of microscopic polyangiitis (MPA); however, the pathogenic mechanisms are not fully understood. Studies have found that circulating microRNA (miRNA) can be used as potential biomarkers for the diagnosis and classification of anti-neutrophil cytoplasmic autoantibody (ANCA)-associated vasculitides (AAV), and the E3 ubiquitin ligase casitas B-lineage lymphoma (CBL) seems to be associated with inflammation. In addition, evidence indicates that miRNA can be tracked into exosomes and transferred into recipient cells to mediate the process of vascular endothelial injury. Herein, we aimed to identify the profiles of exosomal miRNA, and determine the effect of exosomal miR-1287-5p and its target gene CBL on vascular endothelial cells in MPA.

**Method**. We isolated plasma exosomes from patients with MPA (MPA-exo) and healthy controls (HC-exo) by ultracentrifugation and conducted exosome small-RNA sequencing to screen differential miRNA expression in MPA-exo ($n = 3$) compared to HC-exo ($n = 3$). We measured the expression levels of miR-1303, miR-1287-5p, and miR-129-1-3p using quantitative reverse transcription-polymerase chain reaction (qRT-PCR, $n = 6$) and performed dual luciferase reporter gene assays to confirm the downstream target gene of miR-1287-5p. In addition, we treated human umbilical vein endothelial cell (HUVEC) with MPA-exo, or transfected them with miR-1287-5p mimic/inhibitor or with CBL-siRNA/CBL-siRNA+ miR-1287-5p inhibitor. After cell culture, we evaluated the effects on vascular endothelial cells by examining the mRNA levels of IL-6, IL-8, MCP-1, ICAM-1 and E-selectin using qRT-PCR and performed neutrophil adhesion assay with haematoxylin staining.

**Result**. Transmission electron microscopy, Western blot and nanoparticle tracking analysis showed that we successfully purified exosomes and MPA-exo could be absorbed into HUVEC. We screened a total of 1,077 miRNA by sequencing and observed a high abundance of miR-1287-5p in the exosomes obtained from MPA plasma. The dual luciferase reporter assay identified CBL as a downstream target gene of miR-1287-5p, and the results revealed that MPA-exo decreased CBL protein expression in HUVEC. In addition, treatment with MPA-exo, up-regulating miR-1287-5p or silencing of

Corresponding author
Chao Xue, xccqh@126.com

![PeerJ]

CBL in HUVEC significantly increased the mRNA expression of inflammatory factors (including IL-6, IL-8, and MCP-1) and adhesion molecules (including ICAM-1 and E-selection) and promoted the adhesion of neutrophils to HUVEC. However, down-regulating miR-1287-5p had the opposite effect.

**Conclusion**. Our study revealed that MPA-exo was involved in the intercellular transfer of miR-1287-5p and subsequently promote the development of acute endothelial injury in MPA. MiR-1287-5p and CBL agonists may be promising therapeutic approach for MPA-induced vascular inflammatory injury.

# INTRODUCTION

Anti-neutrophil cytoplasmic autoantibody (ANCA)-associated vasculitides (AAV) are disorders characterized by inflammation and destruction of small-sized vessels that lead to endothelial injury and tissue damage accompanied by the presence of ANCA in serum. It is well established that myeloperoxidase (MPO) and proteinase 3 (PR3) are two major target antigens of ANCA (*Nakazawa et al., 2019*). AAV is divided into three clinical phenotypes: granulomatosis with polyangiitis (GPA), microscopic polyangiitis (MPA), and eosinophilic granulomatosis with polyangiitis (EGPA) (*Jennette & Nachman, 2017*). AAV is an uncommon disease with notable ethnicity-based differences, where MPO-ANCA positive MPA predominates in East Asian countries, including China (*Geetha & Jefferson, 2020*). Massive inflammatory injury to vascular endothelial cells can result in necrotizing vasculitis, which is one of the defining histopathological features of MPA. Neutrophils are recognized as effector cells responsible for endothelial damage in acute AAV-related injury (*Al-Hussain et al., 2017*). Neutrophils primed by inflammatory cytokines interact with activated vascular endothelial cells, resulting in tissue inflammation and injury of affected organs through degranulation and the release of free radicals and proteases (*Haubitz, Dhaygude & Woywodt, 2009*). Therefore, considering that MPA is the most common clinical AAV subtype in China, further exploration of the molecular mechanism underlying inflammatory injury of vascular endothelial cells in MPA is particularly crucial to understand the occurrence of acute injury, which will be of great clinical significance reducing mortality.

Exosomes are nanometre-sized microvesicles secreted by almost all mammalian cell types and range in size from 30 to 150 nm. Increasing evidence suggests that exosomes are abundant in biological fluids, including serum, and circulate with the fluids, facilitating communication between various cells and their microenvironment (*Prikryl et al., 2020*; *Surmiak et al., 2021*; *Wang et al., 2020*). Exosomes deliver bioactive molecules, such as mRNA, microRNA (miRNA), long noncoding RNA, proteins, and lipids, from parental cells to recipient cells, playing an important role in many diseases, including cancer, AAV, and acute kidney injury (*Dong et al., 2019*; *Huang et al., 2020*; *Oh & Kwon, 2021*).

MiRNA, which is small noncoding RNA with a length of approximately 20-22 nucleotides, mediates the suppression of target mRNA translation and affects cellular metabolism, proliferation, and inflammatory reactions (*Lara-Barba et al., 2021*). Growing evidence suggests that the quality and quantity of circulating miRNA in AAV patients has changed compared with healthy individuals and is considered as reliable markers for diseases diagnosis and classification (*Skoglund et al., 2015*). A recent study revealed significant correlations between exosomal miR-185-3p and miR-125a-3p and MPA disease activity (*Wang et al., 2020*). In addition, exosomal miRNA derived from neutrophils preincubation with anti-PR3 IgG induces endothelial cells to release adhesion molecules and chemokines. However, the relationship between plasma-derived exosomal miRNA and the inflammatory injury of vascular endothelial cells in MPA remains unclear.

In this study, we investigated the characteristics of miRNA expression profiles in circulating exosomes from MPA patients. We first isolated exosomes from plasma for small-RNA sequencing analysis and then identified the candidate miRNA responsible for MPA by comparing the miRNA expression profiles between MPA patients and healthy controls. MiR-1287-5p was highly expressed in exosomes derived from active MPA. Considering the important role of inflammatory injury of endothelial cell in the pathogenesis of MPA, we speculated that miR-1287-5p from circulating exosomes from active MPA may be absorbed by vascular endothelial cells and contribute to inducing endothelial cell inflammation and neutrophil adhesion.

## MATERIALS & METHODS

### Subjects and sampling

We obtained blood samples from nine MPA patients in the Department of Nephrology of the Second Affiliated Hospital at Guangxi Medical University. All of the MPA patients fulfilled the criteria in the 2012 revised International Chapel Hill Consensus Conference Nomenclature of Vasculitis (*Jennette et al., 2013*) and were positive for IgG anti-MPO antibodies. We excluded patients with serious infections and patients who were positive for anti–PR3 and anti–glomerular basement membrane IgG antibodies. The blood samples were collected before MPA patients received glucocorticoid or immunosuppressants therapy. Nine healthy control individuals (HC) matching the MPA group with respect to age and sex were selected as HC group. The demographic characteristics of the MPA patients and HC in this study are shown in Table 1. The Ethical Committee of the Second Affiliated Hospital at Guangxi Medical University approved this study (Number: KY-0018) in accordance with the Declaration of Helsinki, and we obtained written informed consent from all participants.

### Isolation and purification of exosomes

We purified exosomes from the plasma of HC (HC-exo) and MPA patients (MPA-exo) as previously described with minor changes (*Zhang et al., 2020a*). Briefly, we collected peripheral blood samples in anticoagulant tubes containing EDTA and centrifuged them at $3,000 \times g$ for 15 min at 4 °C to obtain plasma. We then centrifuged the plasma at $12,000 \times g$ for 45 min before filtering it through a 0.22-μm filter. Next, we subjected the collected

**Table 1** Characteristics of study subjects.

| Demographic | MPA | HC |
|---|---|---|
| Number | 9 | 9 |
| Age (y) | 59.22 ± 12.05 | 50.89 ± 6.86 |
| Sex | | |
| Female | 7 | 7 |
| Male | 2 | 2 |
| Hemoglobin (g/L) | 73.22 ± 26.17 | – |
| WBC ($\times 10^9$/L) | 10.95 ± 5.34 | – |
| CRP (mg/L) | 55.05 ± 44.44 | – |
| Creatinine (μmol/L) | 508.67 ± 439.68 | – |
| MPO-ANCA | 246.67 ± 110.00 | – |
| BVAS | 16.89 ± 3.37 | – |

Notes.

MPA, microscopic polyangiitis; HC, healthy control; WBC, white blood cell; CRP, C-reactive protein; BVAS, Birmingham Vasculitis Activity Score.

supernatant to ultracentrifugation at 120,000×g for 70 min at 4 °C. We washed the exosome pallet with ice-cold phosphate-buffered saline (PBS), centrifuged it at 120,000×g for 70 min at 4 °C, and finally resuspended it in 100 μL of PBS.

## Identification of exosomes

After washing the exosome suspension in PBS, we adsorbed the exosomes onto Formvar/carbon support film copper-mesh grids and fixed them with 3% uranyl acetate for negative staining. We observed the ultrastructure of the exosomes using transmission electron microscopy (TEM), detected the expression of exosomal markers *via* Western blot, and analyzed the size distribution of the exosomes *via* nanoparticle tracking analysis (NTA) using a Zetasizer Nano-ZS. We diluted the exosome samples 10,000× with sterilized PBS, injected it into the chamber and quantified the exosomes using NTA software.

## Western blot

We harvested the collected exosomes and the cells with RIPA lysis buffer (Beyotime Biotechnology, Shanghai, China) containing 1% PMSF (a protease inhibitor) for protein extraction and examined the protein concentrations using a BCA protein assay kit (Beyotime Biotechnology, China). We then mixed the proteins in the different groups with loading buffer, denatured the proteins at 100 °C for 5 min and then subjected the proteins to SDS/PAGE. After electrophoresis, we incubated the polyvinylidene difluoride membranes with the following antibodies: anti-CD9 (1:2000, Abmart, Shanghai, China), anti-CD63 (1:2000, Abmart), anti-TSG101 (1:2000, Abmart), anti-CBL (1:2000, Abmart), and anti β-tubulin (1:2000, Abmart). After incubating the membranes overnight at 4 °C, we incubated the membranes with conjugated secondary antibodies (1:10000, Abmart) for 1 h at room temperature. We visualized the images with enhanced chemiluminescence reagent (Epizyme, Shanghai, China) and quantitated them using ImageJ. We compared the grayscale value of each protein signal to that of β-tubulin.

## Culture of HUVEC and treatment with plasma-derived exosomes

We obtained human umbilical vein endothelial cells (HUVEC) from the cell bank of American Type Culture Collection (ATCC) and cultured them at 37 °C with 5% $CO_2$ in endothelial cell medium (ECM) containing 5% foetal bovine serum (FBS), 1% endothelial cell growth supplement, and 100 U/ml of penicillin and streptomycin (HyClone, Logan, UT, USA). We planted the cells with exosomes-depleted FBS and allowed them to reach approximately 90–95% confluence before use. We prepared exosomes-depleted serum by ultracentrifugation at 120,000×g at 4 °C for 18 h followed by passage through a 0.22-μm filter. We treated the HUVEC with HC–exo or MPA–exo for 24 or 48 h. Then, we subjected the treated HUVEC to the following experiments.

## Exosome uptake

To trace the cellular uptake of exosomes, we stained exosomes with the green dye PKH67 (Sigma Aldrich, St. Louis, MO, USA) for 5 min. After terminating the labelling reaction with 5% bovine serum albumin (BSA), we harvested the exosomes and washed them with PBS by centrifugation (110,000×g for 70 min). We resuspended the PKH67–labeled exosomes in FBS–free DMEM and cultured the HUVEC with 50 μg of exosomes for 24 h on confocal dishes at 37 °C with 5% $CO_2$. After incubation, we washed the cells twice with PBS, fixed them with 4% paraformaldehyde for 10 min, and counterstained them with 4′,6-diamidino-2-phenylindole (DAPI) for 5 min to label the nuclei at room temperature. We then observed the cellular uptake of exosomes using a laser scanning confocal microscope (Leica Microsystems, Inc., Wezlar, Germany) and analyzed the absorption efficiency between HC–exo and MPA–exo with ImageJ.

## Transfection

The mimic negative control (MNC), miR-1287-5p mimic, inhibitor negative control (INC) and miR-1287-5p inhibitor were synthesized by GenePharma (Shanghai, China). When the density reached approximately 50–70%, we transfected HUVEC with miR-1287-5p mimic at a concentration of 50 nM and miR-1287-5p inhibitor at a concentration of 100 nM. We used Lipofectamine 3000 (Invitrogen, Carlsbad, CA, USA) as a transfection reagent following the manufacturer's recommendations. After transfection for 6 h, we replaced the antibiotic–free medium with complete medium and cultured the HUVEC for 24 or 48 h. Then, we transfected si–001, si–002, and si–003 (si–CBL) (RiboBio, Guangzhou, China) into HUVEC to determine which siRNA most strongly decreased the mRNA transcription of CBL. We performed the transfection process as described above. Subsequently, we used qRT–PCR or Western blot to evaluate transfection efficiency by examining the mRNA or protein expression of CBL. After determining the most efficient siRNA for CBL silencing, we transfected HUVEC with siRNA negative control (si-NC), si-CBL, or si-CBL combined with miR-1287-5p inhibitor for 24 or 48 h. Table 2 displays the related sequences.

## Exosome small-RNA sequencing and bioinformatics analysis

To conduct small-RNA sequencing, we lysed plasma exosomes isolated from HC ($n = 3$) and MPA patients ($n = 3$) and extracted total RNA using a miRNeasy Micro Kit (Qiagen, Hilden, Germany) according to the manufacturer's instructions. We generated sequencing
**Table 2  Prime sequences for qRT-PCR.**

| Gene | Sequences |
| --- | --- |
| Human IL-6 | Forward: 5′-ACTCACCTCTTCAGAACGAATTG-3′<br>Reverse: 5′-CCATCTTTGGAAGGTTCAGGTTG-3′ |
| Human IL-8 | Forward: 5′-GCCTTCCTGATTTCTGCAGC-3′<br>Reverse: 5′-TGCACTGACATCTAAGTTCTTTAGCA-3′ |
| Human MCP-1 | Forward: 5′-AACTGAAGCTCGCACTCTCG-3′<br>Reverse: 5′-TCAGCACAGATCTCCTTGGC-3′ |
| Human ICAM-1 | Forward: 5′-CAGTCACCTATGGCAACGAC-3′<br>Reverse: 5′-ATTCAGCGTCACCTTGGCTC-3′ |
| Human E-selectin | Forward: 5′-CTCTGACAGAAGAAGCCAA-3′<br>Reverse: 5′-TTGAGTCCACTGAAGCCAGG-3′ |
| Human GAPDH | Forward: 5′-ATTGTTGCCATCAATGACCC-3′<br>Reverse: 5′-AGTAGAGGCAGGGATGATGT-3′ |
| Human miR-1287-5p | 5′-UGCUGGAUCAGUGGUUCGAGUC-3′ |
| Human miR-1303 | 5′-UUUAGAGACGGGGUCUUGCUCU-3′ |
| Human miR-129-1-3p | 5′-AAGCCCUUACCCCAAAAAGUAU-3′ |
| Cel-miR-39 | 5′-UCACCGGGUGUAAAUCAGCUUG-3′ |
| Human miR-1287-5p mimic | Sense: 5′-UGCUGGAUCAGUGGUUCGAGUC-3′<br>Antisense: 5′- CUCGAACCACUGAUCCAGCAUU-3′ |
| Human miR-1287-5p inhibitor | 5′-GACUCGAACCACUGAUCCAGCA-3′ |
| Human si-001 (si-CBL) | 5′-CTACCAGCATCTCCGTACT-3′ |
| Human si-002 (si-CBL) | 5′-GAGCTTTCGACAGGCTCTA-3′ |
| Human si-003 (si-CBL) | 5′-TCGGATTACTAAAGCAGAT-3′ |

libraries using an NEBNext® Multiplex Small RNA Library Prep Set for Illumina® (NEB, Ipswich, MA, USA). We clustered the index-coded samples on a cBot Cluster Generation System using a TruSeq SR Cluster Kit v3-cBot-HS (Illumina) according to the manufacturer's instructions. After cluster generation, we sequenced the library preparations sequenced on an Illumina HiSeq 2500/2000 platform. The Software Novomagic was used to obtain the potential miRNA and volcano map. We considered each miRNA with a $p$ value$<0.05$ and a $\log_2|(\text{fold change})|>2$ to be physiologically relevant. A volcano map showed the differential miRNA expression profiles among exosome samples. The statistical power of experimental design, calculated in RNASeqPower is (depth = 15, cv = 0.4, effect = c (0.8,1.1), alpha = 0.05, power = c(0.8,0.9)).

## RNA extraction and qRT-PCR

We extracted miRNA from the exosomes with a miRNeasy Serum/Plasma Kit (Qiagen, Germany) according to the manufacturer's instructions. We reverse transcribed the extracted miRNA with a miRcute Plus miRNA First-Stand cDNA kit and detected it with a miRcute Plus miRNA SYBR Green qPCR Kit (Tiangen, Beijing, China). We extracted total RNA from HUVEC using TRIzol Reagent (Invitrogen) and performed qRT–PCR with HiScriptII Q RT SuperMix and ChamQ Universal SYBR qPCR Master Mix kits (Vazyme, Shanghai, China). We used an ABI StepOne Real-time PCR instrument (Applied Biosystems, Waltham, MA, U.S.A.) for quantitative analyses. We measured the relative

expression levels of miRNA and mRNA using the $2^{-\Delta\Delta Ct}$ method and standardized them to the expression levels of cel-miR-39 and GAPDH, respectively. Sangon (Shanghai, China) synthesized the primers for miRNA detection, while Vazyme synthesized the primers for mRNA examination. The primer sequences are listed in Table 2.

## Neutrophil adhesion assay

We performed a neutrophil adhesion experiment with minor modifications as previously described (*Choi et al., 2017*; *Nistri et al., 2003*). We isolated neutrophils from the peripheral venous blood of healthy volunteers by using Ficoll-Paque density gradient centrifugation (Tiangen, China) and determined the purity of neutrophils by Giemsa staining. We primed freshly prepared neutrophils ($2 \times 10^5$) with 10 µg/mL lipopolysaccharide (LPS, Sigma Aldrich, USA) for 30 min at 37 °C and added them to HUVEC. We maintained the cocultures for 120 min, rinsed them with PBS three times to remove unbound neutrophils, fixed them in 4% formaldehyde and counterstained them with haematoxylin for light microscopic screening. We performed each group experiment three times and counted the number of adherent neutrophils and HUVEC in five randomly chosen microscopic fields at a 200× final magnification. Finally, we analyzed the ratio of the neutrophils number to HUVEC number.

## Dual-luciferase reporter assay

We predicted miR-1287-5p target genes using the TargetScan database available on the internet (http://www.targetscan.org). We performed a dual-luciferase reporter assay to verify whether *CBL* is the target mRNA of miR-1287-5p. Briefly, we constructed a wild-type *CBL* 3′-untranslated region (3′-UTR) reporter plasmid (*CBL* Wt) and a mutated-type *CBL* 3′-UTR reporter plasmid (*CBL* Mut) with the pMIR-promoter vector by OBiO Technology (Shanghai, China). We plated HEK293T cells in 96-well plates and transfected them with *CBL* Wt or *CBL* Mut in the presence of miR-1287-5p mimic or negative control (NC) in strict accordance to the manufacturer's specifications. At 48 h after transfection, we quantified luciferase activity in the cell lysates with a luciferase assay kit (Promega, Madison, WI, USA).

## Statistical analysis

We performed all statistical analyses using SPSS software and GraphPad Prism 7. We express the data as the mean ± SD. We performed comparisons between two groups using independent-sample $t$-tests. We considered $p < 0.05$ to indicate statistical significance.

# RESULTS

## Characterization and internalization of exosomes derived from plasma

We collected blood samples from both MPA patients and HC. We identified plasma-derived exosomes by TEM, NTA and Western blot. We observed typical cup-shaped vesicles, ∼100 nm in diameter, using TEM (Fig. 1A). We confirmed the expressions of exosomal marker proteins in the two groups, including CD9, CD63 and TSG101, by Western blot analysis (Fig. 1B). NTA showed that the particle size peaks from MPA patients and HC were 80
nm and 100 nm, respectively (Fig. 1C). These results revealed that we successfully isolated exosomes from plasma.

To further determine whether HUVEC absorbed the plasma-derived exosomes, we cocultured exosomes labelled with PKH67 (a green fluorescent cell linker) with HUVEC. After 24 h of incubation, the HUVEC had efficiently internalized PKH67–labeled exosomes, as indicated by confocal laser scanning microscopy (Fig. 1D), which indicated that HUVEC could take up plasma-derived exosomes. Moreover, there was no significant difference between HC–exo and MPA–exo in the average grey value (Fig. 1E).

## Treatment of HUVEC with MPA-exo induced endothelial inflammation and neutrophil adhesion

To investigate whether MPA-exo induced endothelial inflammation, we cocultured HUVEC with HC-exo and MPA-exo for 24 h and measured the expression of inflammatory factors by qRT–PCR. As shown in Fig. 2A, the relative mRNA levels of IL-6, IL-8, and MCP-1 (fold changes: 1.44, 1.82 and 2.07, respectively) and of the adhesion molecules ICAM-1 and E-selectin (fold changes: 1.81 and 1.86, respectively) were significantly higher in HUVEC treated with MPA-exo than in HUVEC treated with HC-exo ($p < 0.05$).

To assess whether MPA-exo attracted neutrophils to the endothelium, we first isolated neutrophils by using Ficoll-Paque density gradient centrifugation. We observed that the cytoplasm was light red, the nuclei was rod-like or 2~5 lobulated, and the purity of cells ranged from 92 to 96% (Fig. 2B). The results indicated that we have successfully isolated neutrophils and the isolated neutrophils could be used for experiments. Then, we cultured HUVEC in the presence of HC-exo or MPA-exo and subsequently incubated the HUVEC monolayer with primed LPS-stimulated neutrophils. As shown in Figs. 2C and 2D, treatment with MPA-exo enhanced the adhesion of neutrophils to HUVEC, and the number of neutrophils adhering to HUVEC was increased three times that in the HC-exo group according to the ratio of neutrophil/HUVCE (HC-exo: $0.12 \pm 0.03$; MPA-exo: $0.24 \pm 0.01$; $p = 0.002$). Our results indicated that MPA-exo treatment significantly increased proinflammatory cytokine release in HUVEC and neutrophil adhesion to HUVEC.

## The miRNA expression profiles of plasma exosomes in MPA patients

To characterize whether the miRNA in MPA-exo play an important role in the development of MPA, we used RNA sequencing to explore the miRNA spectrum between HC and MPA patients. We screened a total of 1,077 miRNA in exosomes purified from plasma samples. According to the criteria of a $p$ value $< 0.05$ and a $\log_2|(\text{fold change})| > 2$, we identified nine differentially expressed miRNA. Of these, six miRNA were upregulated and three miRNA were downregulated in MPA (Table 3). The volcano map displays the overall distribution of differentially expressed miRNA (Fig. 3A). Moreover, we randomly selected three differentially expressed exosomal miRNA (miR-1303, miR-129-1-3p, and miR-1287-5p) for qRT–PCR to validate the sequencing data. We normalized the expression levels of these miRNA to that of cel-miR-39 as a control. The mRNA expressions of miR-1287-5p and miR-1303 were significantly upregulated in MPA samples compared with HC samples, whereas that of miR-129-1-3p was significantly downregulated ($p < 0.05$, Fig. 3B).

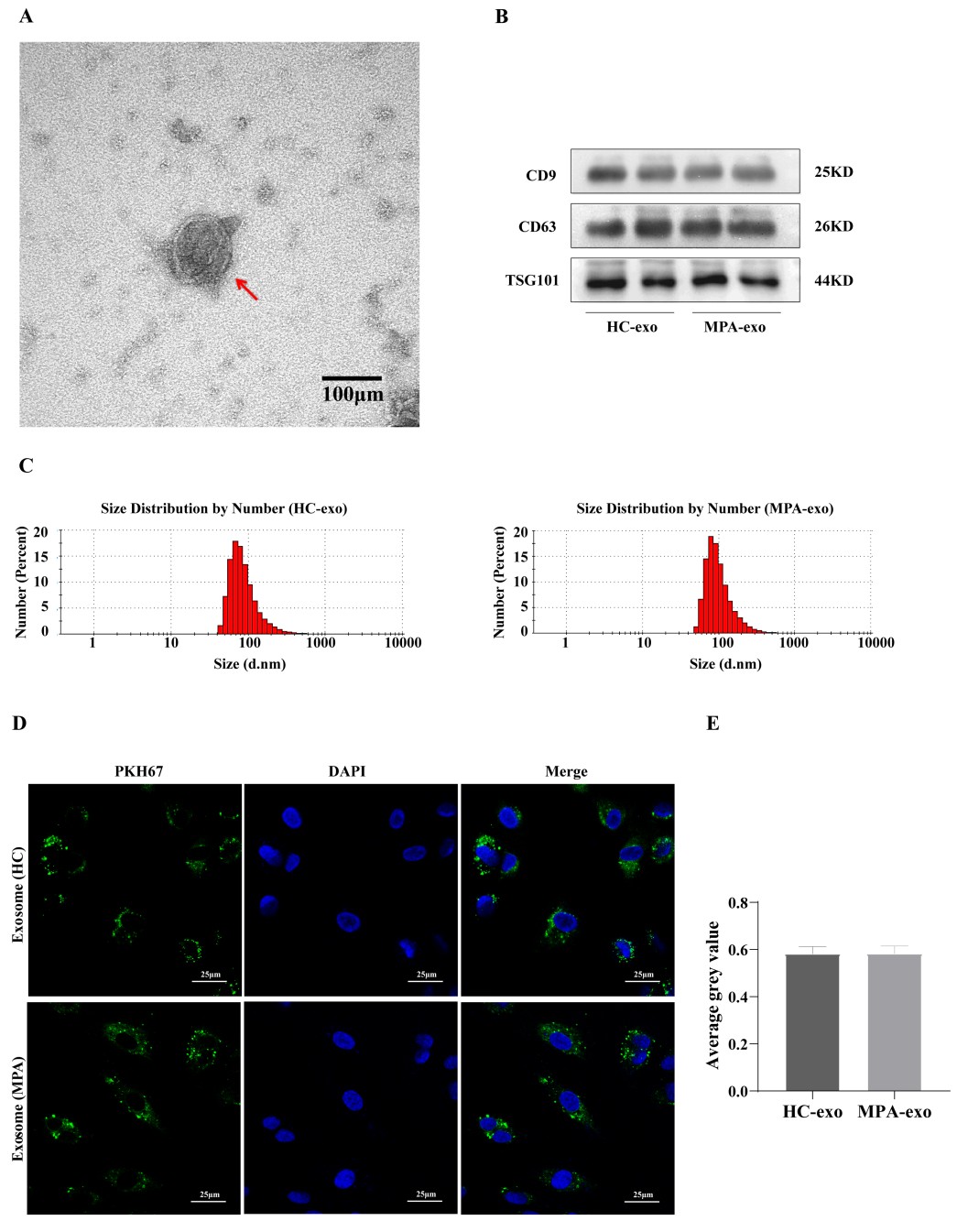

**Figure 1** **Characterization of the plasma-derived exosomes.** (A) Representative micrographs of exosome was visualized by transmission electron microscopy (Scale bar = 100 nm). The red arrow indicates the exosome. (B) Marker proteins CD9, CD63, and TSG101 in exosomes were determined by Western blot. (C) Nanoparticle tracking analysis was used to measure size distribution of exosomes. (D) The uptake of PKH67-labeled exosomes by HUVEC observed using confocal laser 

**Figure 1 (…continued)**
scanning microscopy following co-culture for 24 h (Scale bar = 25 µm). (A) Images of confocal laser scanning microscope. The nuclei of HUVEC were stained with DAPI (blue), the exosome was labeled with PKH67 (green), red arrow indicates the exosomes were localized in the cell nucleus. (E) Showing comparisons of average gray value between HUVEC absorbing exosomes from HC and MPA. Data are presented as mean ± SD and compared with $t$-test. HC-exo, exosomes derived from healthy controls. MPA-exo, exosomes derived from MPA patients; and HUVEC, human umbilical vein endothelial cells.

**Table 3** The expression of differential miRNAs in plasma derived exosomes between MPA patients and healthy controls.

| miRNA | log2 (fold hange) | Up or Down | P value |
|---|---|---|---|
| hsa-miR-223-3p | 2.1 | UP | 0.044 |
| hsa-miR-1303 | 3.1 | UP | 0.044 |
| hsa-miR-1287-5p | 3.9 | UP | 0.029 |
| hsa-miR-3688-5p | 5.0 | UP | 0.034 |
| has-miR-200b-5p | 5.1 | UP | 0.017 |
| has-miR-656-3p | 5.1 | UP | 0.026 |
| hsa-miR-129-1-3p | 5.6 | DOWN | 0.038 |
| Novel_78 | 5.8 | DOWN | 0.030 |
| Novel_310 | 3.6 | DOWN | 0.048 |

## MiR-1287-5p enhanced HUVEC inflammatory response and neutrophil adhesion

For further study, we first examined the basal expression of miR-1303, miR-129-1-3p and miR-1287-5p in HUVEC (Fig. 4A). We found that the basic expression of miR-1303 was relatively high in HUVEC, which indicated that exogenous exosomal miR-1303 had difficulty affecting the endogenous expression level of miR-1303 in HUVEC. The basal expression of miR-129-1-3p in HUVEC and the expression of miR-129-1-3p in plasma-derived exosomes (HC-exo and MPA-exo) were not high, which also suggested that exogenous exosomal miR-129-1-3p and the miR-129-1-3p inhibitor had little influence on the expression of miR-129-1-3p in HUVEC. Therefore, we tested only the changes in miR-1287-5p expression after the uptake of exosomes by HUVEC after 24 h, and found that the expression levels of miR-1287-5p increased in the HUVEC ($p < 0.05$, Fig. 4B). Subsequently, we determined the effects of miR-1287-5p on HUVEC's inflammatory response and neutrophil adhesion. The qRT–PCR results shown in Fig. 4C suggested that the miR-1287-5p mimic predominantly increased the mRNA expression of IL-6, IL-8 and MCP-1 (fold changes: 2.29, 1.99, and 2.50, respectively), while the miR-1287-5p inhibitor significantly inhibited the mRNA expression of these factors compared to that in the control group (fold changes: 0.50, 0.36 and 0.31, respectively, ($p < 0.05$); we obtained marginal evidence for MCP-1 ($p = 0.081$). We also investigated the mRNA expression of adhesion molecules by qRT–PCR. In comparison with the control group, ICAM-1 and E-selectin mRNA levels were markedly elevated after treatment with miR-1287-5p mimic (fold changes: 2.36 and 1.17), but were reduced after treatment with the miR-1287-5p inhibitor (fold changes: 0.53 and 0.26, ($p < 0.05$, Fig. 4C).

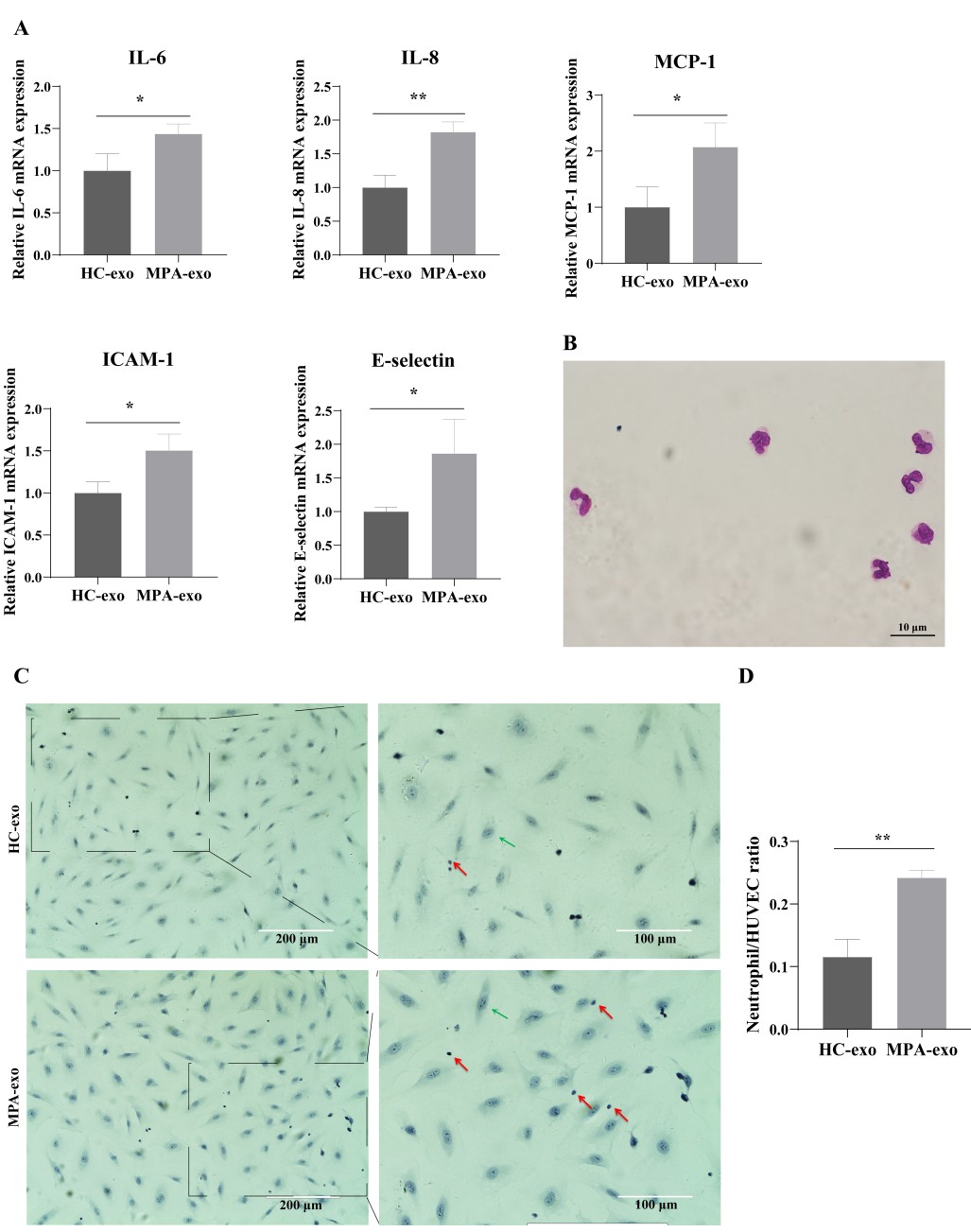

**Figure 2** **Plasma-derived exosomes from the MPA patients induced inflammatory response in HU-VEC.** HUVEC were treated with HC-exo or MPA-exo for 24 h. (A) The inflammatory factors and adhesion molecules IL-6, IL-8, MCP-1, ICAM-1, and E-selectin levels were detected by qRT-PCR after co-culture with MPA-exo or HC-exo. (B) Neutrophils were identified by Giemasa stain (Scar bar = 10 μm). (C) Representative light micrographs of neutrophils adhering to HUVEC. Nuclei were counterstained with hematoxylin. The red arrow indicates adherent neutrophils, and the green arrow indicates HUVEC. Scale bar = 200 μm & 100 μm. 

**Figure 2 (…continued)**
(D) Bar chart showing the neutrophil/HUVEC ratio. The numbers of adherent neutrophils and HUVEC were counted in five randomly chosen microscopic fields at a 200 × final magnification. Data are presented as mean ± SD and compared with $t$-test. ** $p < 0.01$. HC-exo, exosomes derived from healthy controls; MPA-exo, exosomes derived from MPA patients; and HUVEC, human umbilical vein endothelial cells.

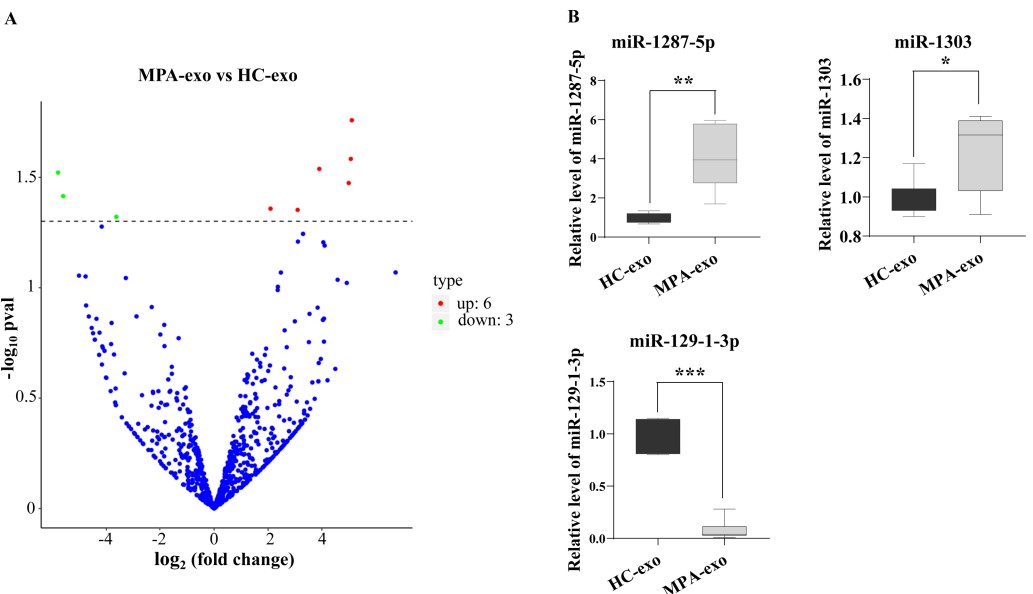

**Figure 3** **RNA-seq identified the differential expression of miRNA in MPA-exo.** (A) Volcanic map revealed differentially expressed miRNA in HC-exo or MPA-exo ($n = 3$). The $x$-axis is $\log_2$ (fold change) and the $y$-axis is $-\log_{10}$ ($P$ value). Blue dots represent miRNA with no significant differences, red dots represent up-regulated miRNA and green dots represent down-regulated miRNA. (B) The expression profiles of the randomized selected three differential exosomal miRNA were validated by qRT-PCR ($n = 6$). Data are presented as mean ± SD and compared with $t$-test. * $p < 0.05$, and ** $p < 0.01$. HC-exo, exosomes derived from healthy controls; and MPA-exo, exosomes derived from MPA patients.

The neutrophil adhesion assay results shown in Fig. 5A revealed that miR-1287-5p mimic promoted the adhesion of primed neutrophils to the HUVEC, which was suppressed by the miR-1287-5p inhibitor (Fig. 5B, ($p < 0.05$). Similarly, compared with MNC group, the number of neutrophils adhering to HUVEC increased 2.36 times in miR-1287-5p mimic group according to the ratio of neutrophil/HUVEC (MNC: $0.5 \pm 0.14$; mimic: $1.18 \pm 0.14$; $p = 0.004$); while the ratio of neutrophil/HUVEC in miR-1287-5p inhibitor group was decreased in contrast to INC group (INC: $0.53 \pm 0.12$; inhibitor: $0.31 \pm 0.04$, $p = 0.036$).

## CBL is a target gene of miR-1287-5p

We next investigated the downstream mechanism by which plasma-derived exosomal miR-1287-5p induces HUVEC inflammatory response and neutrophil adhesion. We predicted a binding site in CBL-mRNA for miR-1287-5p using TargetScan (Fig. 6A). We constructed *CBL* Wt and *CBL* Mut in OBiO Technology (Shanghai, China, Fig. 6B) and used them to transfect HEK393T cells together with miR-1287-5p mimic or MNC. The

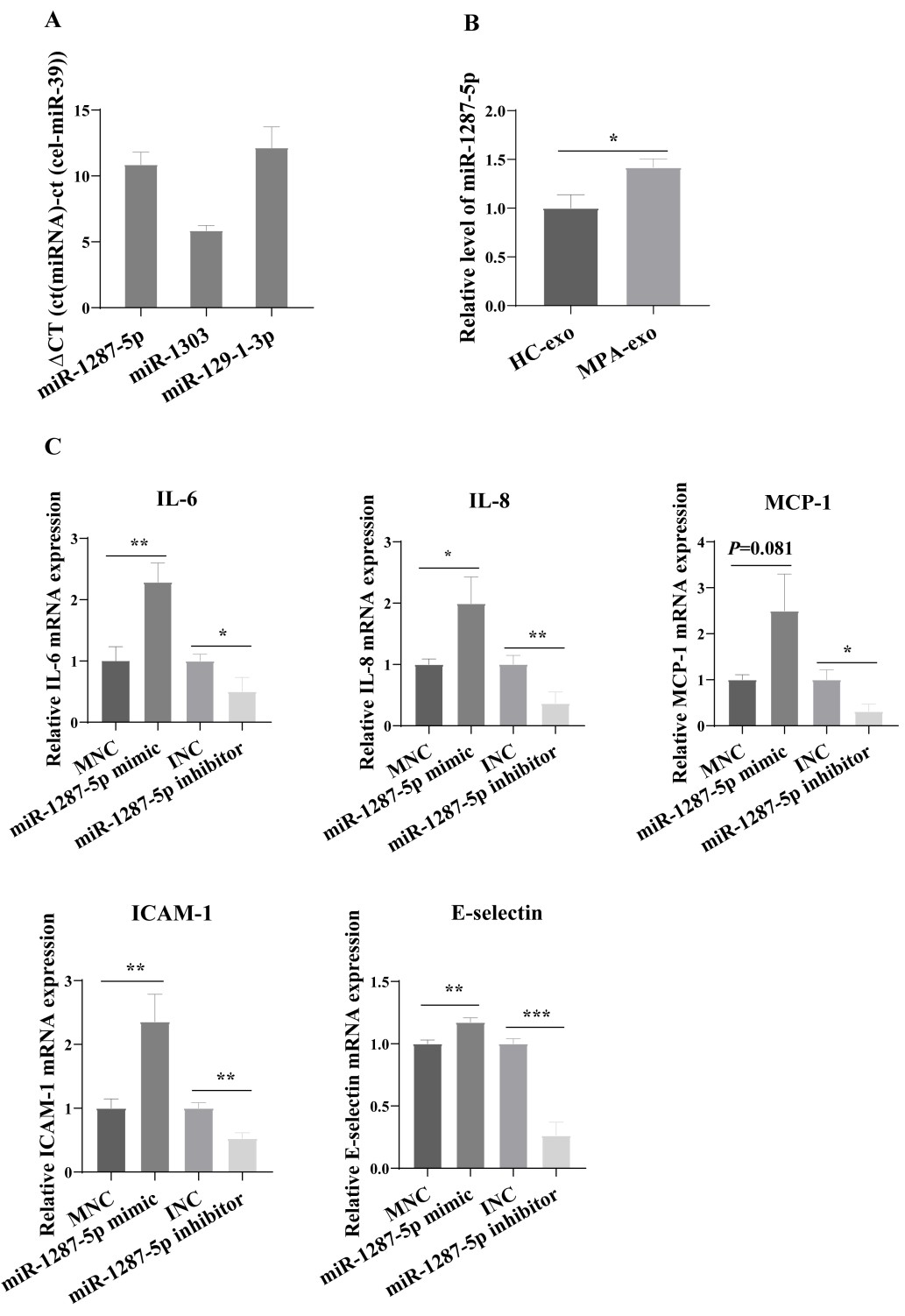

**Figure 4** **MiR-1287-5p induced inflammatory response in HUVEC.** (A) The basal expression levels of miR-1287-5p, miR-1303, and miR-129-1-3p in HUVEC were detected by qRT-PCR. Delta CT was used to compare the levels of miRNA to the cel-miR-39. 

**Figure 4 (…continued)**
(B) The expression of miR-1287-5p in HUVEC was detected by qRT-PCR after treated with MPA-exo for 24 h. (C) Effect of transfecting HUVEC with miR-1287-5p mimic and inhibitor on inflammatory factors and adhesion molecules. Data are presented as mean ± SD from three independent experiments and compared with $t$-test. * $p < 0.05$, ** $p < 0.01$ and *** $p < 0.001$. HC-exo, exosomes derived from healthy controls; MPA-exo, exosomes derived from MPA patients; MNC, mimic negative control; and INC, miRNA inhibitor negative control.

results of the dual luciferase reporter assay suggested that cotransfection with miR-1287-5p mimic decreased the luciferase activity of *CBL* Wt ($p < 0.001$), but not that of *CBL* Mut ($p = 0.0031$, Fig. 6C). Subsequently, we determined the *CBL*-mRNA levels using qRT–PCR after transfecting HUVEC with a miR-1287-5p mimic or a miR-1287-5p inhibitor for 24 h. As shown in Fig. 6D, *CBL*-mRNA was downregulated in HUVEC treated with the miR-1287-5p mimic ($p < 0.05$). The *CBL*-mRNA level in HUVEC increased by 1.35 times after transfection with the miR-1287-5p inhibitor, but the difference was not significant ($p = 0.058$). Western blot analysis also demonstrated that HUVEC treated with miR-1287-5p mimic for 48 h exhibited marked reductions in CBL protein expression; however, HUVEC treated with miR-1287-5p inhibitor exhibited upregulation of CBL protein expression (Fig. 6E). In brief, these results indicated that *CBL*-3′ UTR-mRNA contains a binding site targeted by miR-1287-5p.

## Downregulation of CBL promoted inflammatory response and neutrophil adhesion in HUVEC

To determine whether MPA-exo affect CBL protein expression in HUVEC, we cultured HUVEC in the presence of NC-exo or MPA-exo for 48 h. As shown in Fig. 7A, treatment with MPA-exo decreased CBL protein expression in HUVEC ($p < 0.05$). Subsequently, to explore the effect of *CBL* gene on inflammation and neutrophil adhesion in HUVEC, we used siRNA (si-001, si-002, si-003) to silence *CBL*-mRNA expression. Given the results of both qRT–PCR and Western blot, the most efficient siRNA for *CBL* silencing was si-003 (Figs. 7B&7C). HUVEC transfected with si-CBL (si-003) exhibited significantly increased the mRNA expression of IL-6, IL-8, MCP-1, ICAM-1 and E-selectin after 48 h compared with that in the si-NC group (fold changes: 1.44, 3.44, 2.14, 3.87, and 5.03, respectively, $p < 0.05$, Fig. 7D). Blocking CBL protein expression also contributed to neutrophil adhesion on HUVEC, the ratio of neutrophil/HUVEC was increased in si-003 group compared with si-NC group (si-NC: 0.55 ± 0.06; si-003: 1.66 ± 0.14; $p < 0.05$, Figs. 8A&8B). In addition, the dysfunction induced by si-CBL was reversed by the miR-1287-5p inhibitor. Compared with si-003 group, the mRNA expression of IL-6, IL-8, MCP-1, ICAM-1 and E-selectin was decreased in si-003+miR-1287-5p inhibitor group (fold changes: 0.74, 0.43, 0.71, 0.39, and 0.28, respectively, $p < 0.05$). The ratio of neutrophil/HUVEC in si-003 group and si-003+miR-1287-5p inhibitor group was 1.66 and 0.48, respectively. Taken together, the results revealed that CBL protein inhibition was associated with an increased inflammatory response and increased neutrophil adhesion.

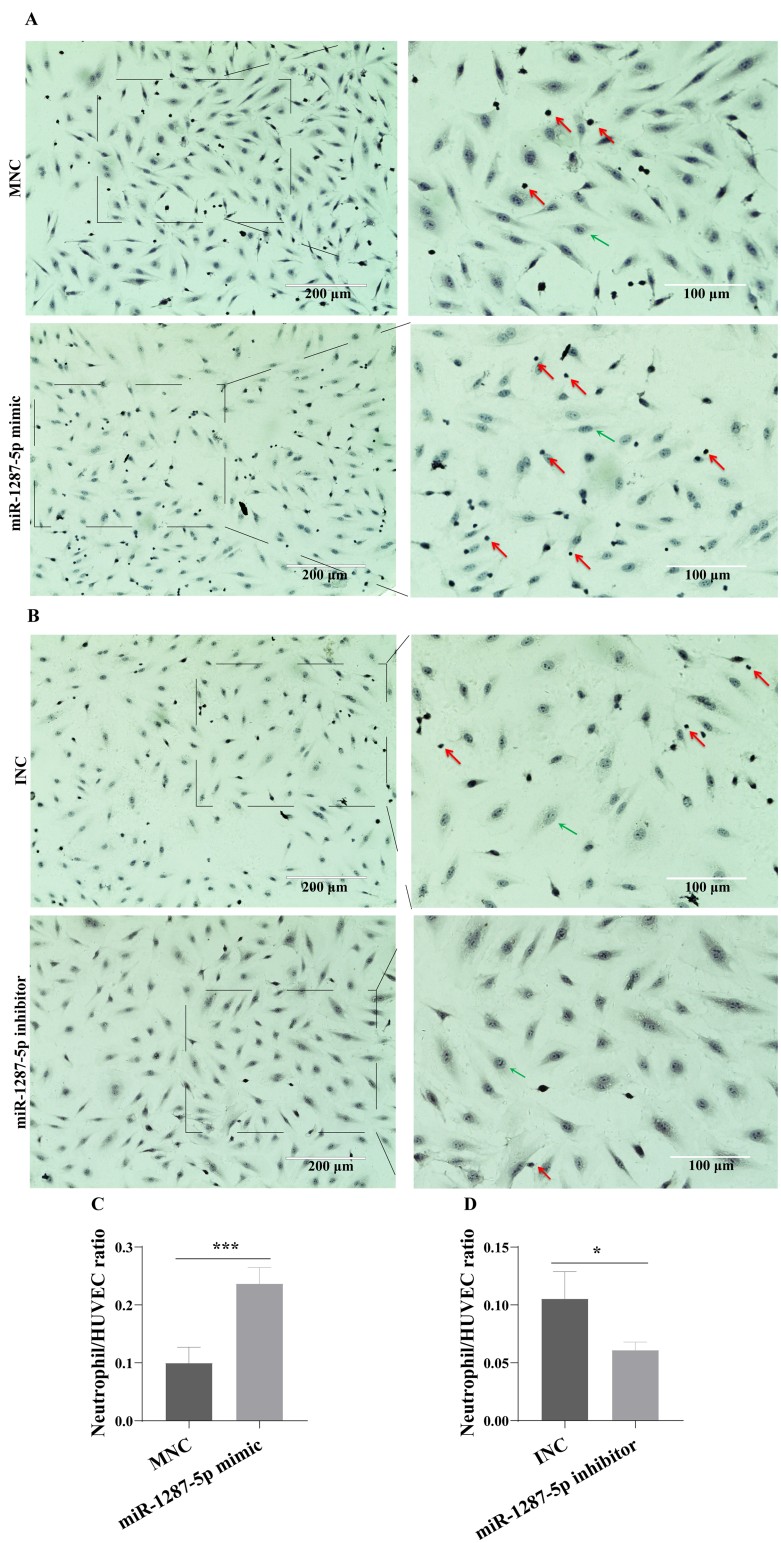

**Figure 5  MiR-1287-5p promoted neutrophils adhesion to HUVEC.** HUVEC were transfected with miR-1287-5p mimic, inhibitor, and negative control for 48 h. (continued on next page...)

**Figure 5 (...continued)**
(A&B) Representative light micrographs of neutrophils adhering to HUVEC after transfected with miR-1287-5p mimic and MNC. (C&D) Representative light micrographs of neutrophils adhering to HUVEC after transfected with miR-1287-5p inhibitor and INC. Nuclei were counterstained with hematoxylin. The red arrow indicates adherent neutrophils, and the green arrow indicates HUVEC. Scale bar = 200 μm & 100 μm. (D) Bar chart showing the neutrophil/HUVEC ratio. The numbers of adherent neutrophils and HUVEC were counted in five randomly chosen microscopic fields at a 200 × final magnification. Data are presented as mean ± SD from three independent experiments and compared with $t$-test. $^{*}$ $p < 0.05$, and $^{***}$ $p < 0.001$. HC-exo, exosomes derived from healthy controls; MPA-exo, exosomes derived from MPA patients; MNC, mimic negative control; and INC, miRNA inhibitor negative control.

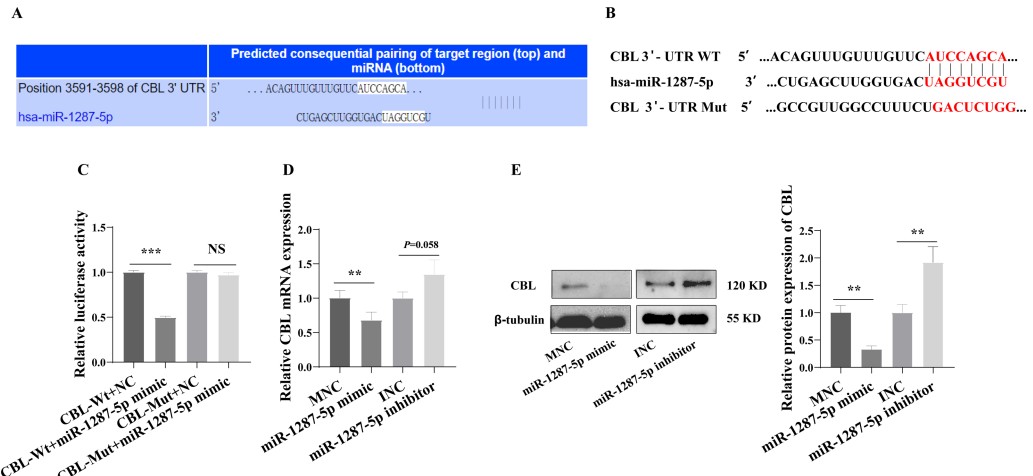

**Figure 6** **CBL is a direct target of miR-1287-5p.** (A) TargetScan software predicted that CBL was a potential target of miR-1287-5p. (B) Binding site of CBL mRNA 3′-UTR (CBL-Wt) with miR-1287-5p and designed CBL 3′-UTR mutant (CBL-Mut) sequence. (C) Luciferase activity was significantly decreased in 293T cells cotransfected with the CBL-Wt vector and miR-1287-5p mimic but was no significance difference in cells cotransfected with CBL-Mut and miR-1287-5p mimic, when compared to the negative control. (D) qRT-PCR and (E) Western blot analysis was used to detect the mRNA and protein expression of CBL in HUVEC transfected with miR-1287-5p mimic, inhibitor, or negative control for 24 or 48 h. Data are presented as mean ± SD from three independent experiments and compared with $t$-test. $^{**}$ $p < 0.01$, and $^{***}$ $p < 0.001$. MNC, mimic negative control; INC, miRNA inhibitor negative control; and NS, no significance.

# DISCUSSION

The pathophysiological mechanisms associated with MPA are complex and are not entirely understood. An increasing number of studies have confirmed an inflammatory damage to the vascular endothelium caused by leukocyte infiltration, a pathological hallmark of MPA (*Nagao et al., 2011*; *Rymarz, Mosakowska & Niemczyk, 2021*). In the present study, we aimed to determine whether circulating exosomes from MPA are involved in vascular endothelial inflammatory damage and to explore the potential mechanisms. The results showed that MPA-exo increased the release of inflammatory factors and adhesion molecules in HUVEC, and promoted the adhesion of neutrophils to HUVEC. Furthermore, they suggested that exosomal miR-1287-5p might regulate vascular endothelial inflammatory damage by targeting CBL.

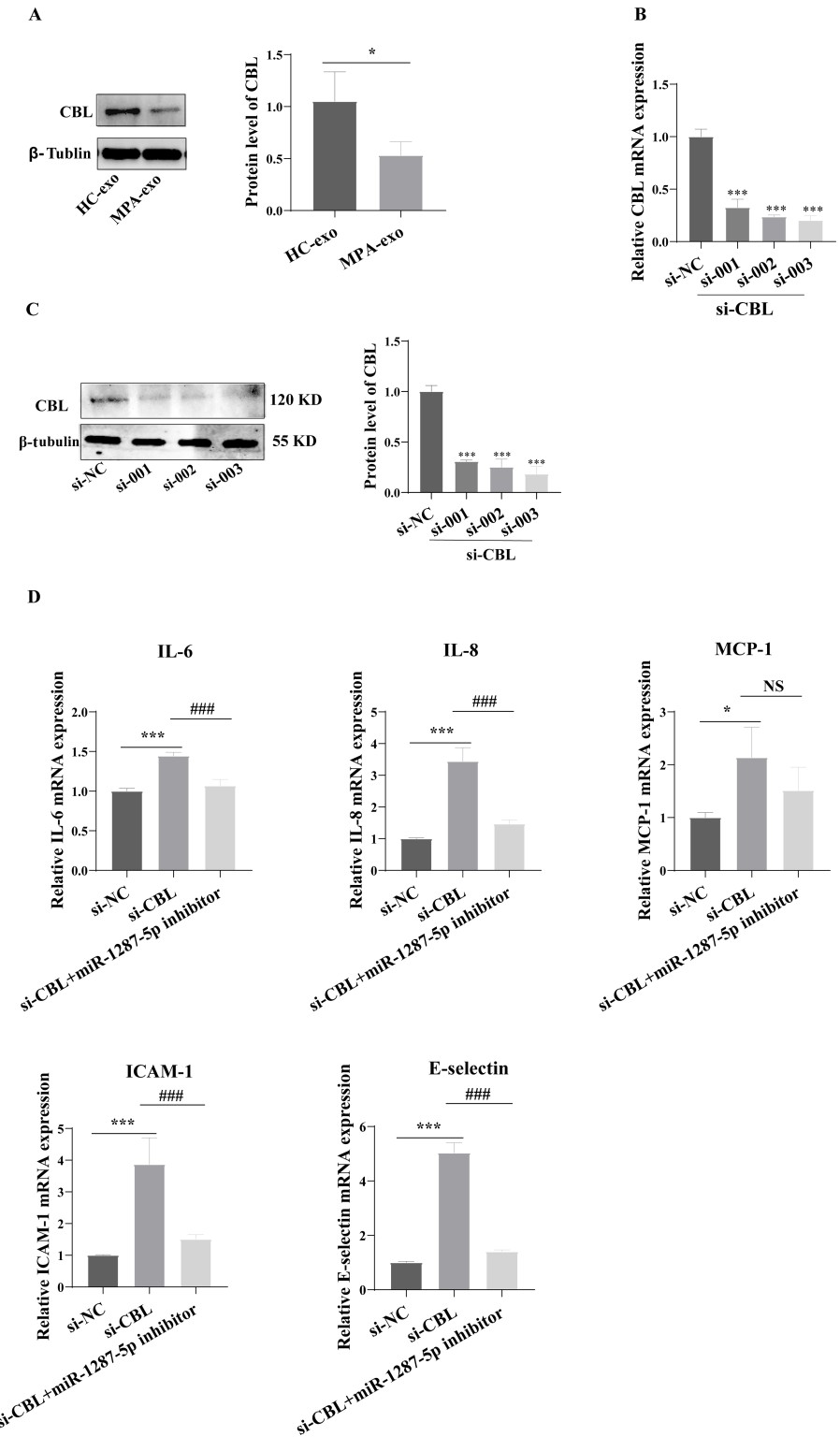

**Figure 7 Silencing CBL induced inflammatory response in HUVEC.** (A) CBL protein level in HUVEC was measured using Western blot after treated with HC-exo or 

**Figure 7 (…continued)**
MPA-exo for 48 h. (B) HUVEC was transfected with CBL siRNA (si-NC, si-001, si-002, and si-003) for 48 h, and mRNA expression of CBL was evaluated by RT-PCR to select the most efficient siRNA. (C) CBL protein level in HUVEC was measured using Western blot after siRNA transfection. (D) The effect of si-CBL or/and miR-1287-5p inhibitor treatment on HUVEC inflammatory factors and adhesion molecules. Data are presented as mean ± SD from three independent experiments and compared with $t$-test. * $p <$ 0.05, *** $p < 0.001$, and si-NC compared with si-CBL; ### $p < 0.001$, and si-CBL compared with si-CBL + miR-1287-5p inhibitor. HC-exo, exosomes derived from healthy controls; and MPA-exo, exosomes derived from MPA patients; si-NC, negative control siRNA; and NS, no significance.

Exosomes are potent intercellular communicators that mediate cell injury or repair by carrying certain cargo (such as miRNA) into recipient cells and have attracted extensive research attention in recent years (*Xu, Jia & Xu, 2019*). Previous studies have indicated that exosomes participate in immune modulation and are related to the immune pathogenesis of autoimmune diseases, such as systemic lupus erythematosus (SLE) and rheumatoid arthritis (RA) (*Tan et al., 2016*). Further investigations have shown that exosomal miRNA participate in the progression of various diseases by affecting the inflammatory response (*Miao et al., 2021*; *Wang et al., 2021*). Exosomes are cup-shaped membrane vesicles with a diameter between 30 and 150 nm that are visible under electron microscopy. Researchers can detect the unique proteins exposed to exosomal membranes to further characterize exosomes. Exosomes from different sources can express proteins that participated in multivesicular body biogenesis (such as Alix and TSG101), integrins and tetraspanins (such as CD9, CD63, CD81 and CD82) (*Shao et al., 2018*). In this study, the extracellular vesicles we isolated from plasma were cup-shaped membrane vesicles with particle size peaks of 80 nm and 100 nm that contained unique exosomal proteins, including CD9, CD63 and TSG101. The findings revealed that we successfully purified exosomes from the plasma of MPA patients and HC. Currently, the commonly used exosome isolation methods include ultracentrifugation, gradient ultracentrifugation, and commercial assays. Royo et al. analysed the expression of exosomal miRNA using miRNA profiling and found that the approach use to isolate exosomes had little effect on exosome-associated miRNA (*Royo et al., 2016*). To reduce the influences of the solvent in the commercial kits on the subsequent cell experiments, we used the most commonly conventional ultracentrifugation method for exosome isolation.

Exosomal entry into recipient cells occurs *via* a variety of mechanisms, such as endocytosis/phagocytosis, fusion, and receptor/ligand signalling (*McKelvey et al., 2015*; *Svensson et al., 2013*). Similar to other types of cells, vascular endothelial cells have the ability to produce and take up exosomes. Therefore, circulating exosomes may be captured by vascular endothelial cells throughout the body and cause cell injury, which may contribute to the various pathophysiologies of organs in the above-described diseases and MPA (*Gao et al., 2021*; *Roig-Carles et al., 2021*; *Wang et al., 2020*). In the current study, we observed that exosomes isolated from the plasma of MPA patients and HC could be absorbed into HUVEC *via* endocytosis. The results further showed that MPA-derived exosomes induced an inflammatory response in HUVEC with neutrophil adhesion. To fully explore the molecular mechanism by which MPA-exo promote inflammation, we

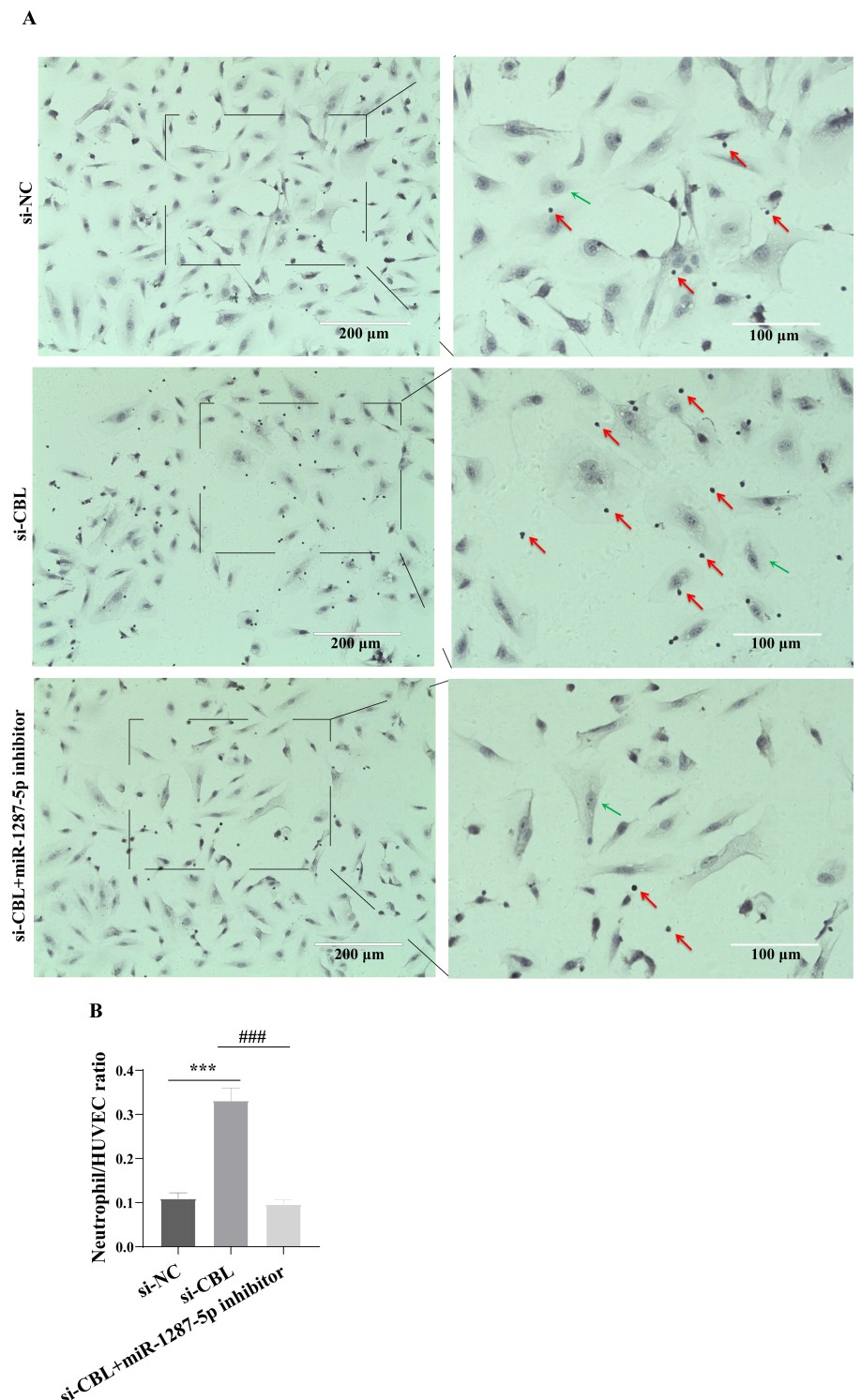

**Figure 8** **Silencing CBL promoted neutrophils adhesion to HUVEC.** (A) The effect of si-CBL or/and miR-1287-5p inhibitor treatment on HUVEC neutrophils adhesion, (continued on next page…)

**Figure 8 (…continued)**
Nuclei were counterstained with hematoxylin. The red arrow indicates adherent neutrophils, and the green arrow indicates HUVEC. Scale bar = 200 µm & 100 µm. (B) Bar chart showing the neutrophil/HUVEC ratio. The numbers of adherent neutrophils and HUVEC were counted in five randomly chosen microscopic fields at a 200 × final magnification. Data are presented as mean ± SD from three independent experiments and compared with $t$-test. *** $p < 0.001$, and si-NC compared with si-CBL; ### $p < 0.001$, and si-CBL compared with si-CBL + miR-1287-5p inhibitor. si-NC, negative control siRNA.

performed RNA sequencing to identify the exosomal miRNA profile. Overall, we screened a total of 1,077 miRNA, of which six miRNA were significantly upregulated and three miRNA were significantly downregulated in MPA. As discussed in a previous study, our work did not produce the same differential miRNA profiles as those reported by Wang et al. (*Wang et al., 2020*). This could be attributable to minor technical differences or sample storage procedures. *Te et al. (2010)* reported that miRNA profiles are differentially expressed in lupus nephritis in different racial populations. Furthermore, we identified by qRT–PCR that miR-1287-5p was significantly enriched in MPA plasma–derived exosomes compared exosomes from HC. We showed a significant uptake of exosomal miR-1287-5p by HUCEC. The results might indirectly indicate that plasma exosomes in MPA may transfer pro-inflammatory miRNA such as miR-1287-5p to the site of vasculitis. A previous review has reported that exosomes play a critical role in the development of autoimmune vasculitis (*Wu et al., 2019*). In addition, circulating exosome levels are correlated with disease activity in SLE (*Lee et al., 2016*). Therefore, we hypothesize that circulating exosomes might be absorbed by endothelial cells in MPA and participate in vasculitis progression through a proinflammatory effect of miR-1287-5p.

MiR-1287-5p was first identified in tumors as a significant suppressor in different cancer, including breast cancer (*Schwarzenbacher et al., 2019*), Cervical Cancer (*Ji et al., 2020*), pancreatic cancer (*Zhang et al., 2020b*), and non-small cell lung cancer (*Shanshan et al., 2021*). MiR-1287-5p regulates cell growth, apoptosis, ferroptosis, and invasion by targeting phosphoinositide 3-kinase CB, glutathione peroxidase 4, and Y-box binding protein 1 (*Cui, Zhao & Li, 2020*; *Schwarzenbacher et al., 2019*; *Shanshan et al., 2021*). Recent studies have proposed that miR-1287-5p also participates in other biological functions. Transfecting miR-1287-5p–mimic,increases caspase-3 activity and inhibits angiogenesis in HUVEC by targeting angiopoietin-1 (*Sanchez et al., 2019*). According to Zhang et al. miR-1287-5p is a target of circ_0004812, and participates in HBV-induced immune suppression by targeting follistatin-related protein (FSTL1) (*Zhang & Wang, 2020*). In a late diabetic kidney disease cohort, Satake et al. demonstrated that circulating miR-1287-5p was strongly associated with the 10-year risk of end-stage kidney disease. In addition, miR-1287-5p mimic indirectly upregulated the concentrations of EPH receptor A2 (EPHA2) in HUVEC lysate, supernatants, or both by targeting CBL (*Satake et al., 2021*). EPHA2 activation has been shown to induce proinflammatory gene expression and stimulate monocyte adhesion in endothelial cells, which indicates that miR-287-5p plays a role in promoting the endothelial cell inflammatory response (*Funk et al., 2012*). Based on these findings, we speculated that miR-1287-5p carried by plasma-derived exosomes is related to the impact of MPA on vascular inflammation. The results of our study may support this hypothesis.

We observed elevated production of proinflammatory genes, activation of endothelial cells and increased adhesion of neutrophils to HUVEC in the miR-1287-5p mimic group; conversely, these effects were reversed with miR-1287-5p inhibitor treatment. The results of this study as well as previous research have demonstrated that the microenvironment of exosomal parent cells affects the cargo and function of exosomes and that exosomes derived from pathological conditions may mediate body injury (*Chang et al., 2018*).

MiRNA induce posttranscriptional gene silencing by binding to the 3′ UTR of target genes and mostly interrupting mRNA expression and/or inhibiting protein synthesis. To elucidate the molecular mechanism by which miR-1287-5p influences the HUVEC inflammatory response, we predicted possible target genes of miR-1287-5p using the TargetScan algorithm. Among the targets, CBL was reported to be associated with the inflammatory response (*Duan et al., 2021*), so we selected CBL as the potential target of miR-1287-5p. The dual-luciferase reporter assay supported this hypothesis, as miR-1287-5p reduced the activity of the luciferase/CBL-3′UTR construct. In addition, Western blot indicated that the miR-1287-5p mimic downregulated CBL protein expression in HUVEC, while the miR-1287-5p inhibitor enhanced CBL expression. Previous research has identified CBL, a E3 ubiquitin ligase, as a negative regulator of receptor tyrosine kinase (RTK) signalling, and the function of CBL is dependent on the ubiquitylation of active RTKs (*Wang et al., 2002*). CBL participates in controlling immature helper T-cell development, immune tolerance, and angiogenesis. It has been reported that CBL might alleviate endothelial dysfunction in patients with diabetes mellitus by inactivating the JAK2/STAT4 signalling pathway (*Jin et al., 2021*). In addition, a study by Duan et al. demonstrated the role of CBL in regulating intestinal inflammation *via* inhibition of fungus-induced noncanonical NF-$\kappa$B activation (*Duan et al., 2021*). The above findings reveal the endothelial protective function and anti-inflammatory function of CBL. In addition, Chiou et al. reported that CBL is a critical mediator in Toll-like receptor stimulation, which plays a key role in the pathogenesis of AAV (*O'Sullivan et al., 2018*). Here, we showed that CBL expression was decreased in HUVEC when exposed to MPA-exo. We also found that the reduction in CBL expression caused by transfection of HUVEC with siRNA had an effect as similar to that of the miR-1287-5p mimic on the function of HUVEC. These findings suggest that CBL deficiency promotes the inflammatory response and neutrophil adhesion of HUVEC, which are pathological features of MPA. Alveolar capillary inflammation can cause pulmonary haemorrhage followed by hypoxia (*Hayes-Bradley, 2004*). Hypoxia is a typical microenvironmental feature of most autoimmune diseases, affecting a variety of physiological processes (*Deng et al., 2022*; *Li et al., 2022b*; *Lohrasbi et al., 2022*). One study has shown that sustained hypoxia stress can repress the CBL expression and disrupt ubiquitin degradation, leading to impaired regulation of human trophoblasts, which is the main mechanism of preeclampsia (*Li et al., 2022a*). However, it has also been shown that hypoxia induces an increase in CBL expression in H9c2 cells. Thus far, there have been few studies on the impacts of hypoxia on MPA, and the relationship between hypoxia and CBL in MPA also needs further exploration.

Nevertheless, this study encountered a limitation. We only measured cytokine mRNA and not protein that promotes the inflammatory environment. It is well known that mRNA

would be translated to protein. However, we did not prove this and further investigation is expected.

## CONCLUSIONS

The current study showed that miR-1287-5p was significantly enriched in MPA exosomes. In a HUVEC cell model, it mediated the expression of inflammatory factors and adhesion molecules, and promoted neutrophil adhesion to HUVEC by targeting CBL, which was associated with a vascular inflammatory environment. Thus, miR-1287-5p might contribute to the acute injury in MPA. As an E3 ubiquitin ligase, CBL is involved in inflammation and immune regulation. In this study, MPA-exo reduced CBL protein expression, and CBL deficiency induced an inflammatory response in HUVEC. Whether CBL agonists can protect against inflammatory injury of endothelial cells and play a therapeutic role in AAV warrants further investigation. In summary, our findings demonstrate, for the first time, that miR-1287-5p target *CBL* and may provide a potential new target for the treatment of MPA-induced vascular inflammatory injury.

## ACKNOWLEDGEMENTS

We thank our colleagues in the Department of Nephrology, the Second Affiliated Hospital of Guangxi Medical University and the Experimental Center of Guangxi Medical University for their administrative and academic support. We also thank all participants enrolled in this study.

### Funding

This work was supported by funding from the Guangxi Natural Science Foundation Program (No. 2018GXNSFAA281122), the Development and Application Project of Medical and Health in Guangxi Zhuang Autonomous Region (No. S2017010), the NSFC cultivation project of The Second Affiliated Hospital of Guangxi Medical University (No. GJPY2018009), and the Health and Family Planning Commission of Hunan Province (No. 202203054404). The funders had no role in study design, data collection and analysis, decision to publish, or preparation of the manuscript.

### Grant Disclosures

The following grant information was disclosed by the authors:
The Guangxi Natural Science Foundation Program: 2018GXNSFAA281122.
The Development and Application Project of Medical and Health in Guangxi Zhuang Autonomous Region: S2017010.
The NSFC cultivation project of The Second Affiliated Hospital of Guangxi Medical University: GJPY2018009.
The Health and Family Planning Commission of Hunan Province: 202203054404.

## Competing Interests

The authors declare there are no competing interests.

## Author Contributions

- Yan Zhu conceived and designed the experiments, performed the experiments, prepared figures and/or tables, authored or reviewed drafts of the article, and approved the final draft.
- Liu Liu performed the experiments, prepared figures and/or tables, authored or reviewed drafts of the article, and approved the final draft.
- Liepeng Chu analyzed the data, prepared figures and/or tables, authored or reviewed drafts of the article, and approved the final draft.
- Jingjing Lan performed the experiments, prepared figures and/or tables, and approved the final draft.
- Jingsi Wei analyzed the data, prepared figures and/or tables, authored or reviewed drafts of the article, and approved the final draft.
- Wei Li analyzed the data, authored or reviewed drafts of the article, and approved the final draft.
- Chao Xue conceived and designed the experiments, prepared figures and/or tables, authored or reviewed drafts of the article, and approved the final draft.

## Human Ethics

The following information was supplied relating to ethical approvals (i.e., approving body and any reference numbers):

This study was approved by the Ethical Committee of the Second Affiliated Hospital at Guangxi Medical University (Number: KY-0018) in accordance with the Declaration of Helsinki.

## Microarray Data Deposition

The following information was supplied regarding the deposition of microarray data:

Title: Plasma-derived exosome miRNA sequencing between patients with microscopic polyangiitis and healthy controls

Accession number: PRJNA814192

## Data Availability

The raw measurements are available in the Supplemental Files.

## Supplemental Information

Supplemental information for this article can be found online at http://dx.doi.org/10.7717/peerj.14579#supplemental-information.

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
