# Peer review of "Microscopic polyangiitis plasma-derived exosomal miR-1287-5p induces endothelial inflammatory injury and neutrophil adhesion by targeting CBL"

_PeerJ, doi:10.7717/peerj.14579_

## Round 0.1 · original submission · Major Revisions

Although both reviewers find the study interesting, they raised multiple concerns related to the descriptions of the findings and data interpretations. Please address the comments provided by the reviewers. Especially, the second reviewer provided informative comments to further improve the manuscript. Please make sure that there are no grammatical or typographical errors.

Reviewer 1 ·

Basic reporting

You have demonstrated that miR-1287-5p targeting CBL gene in MPA-derived exosomes mediates vascular inflammation and a neutrophil adhesion promotion on HUVECs for the first time, and speculated miR-1287-5p and/or CBL could be a treatment targeted for the MPA.

Some minor points to be addressed.
1. There are several difficulty to read your article. Please proofread your article by a fluent speaker.
2. Please respect submission guidelines and scientific notation (space between number and units, italics on "et al" etc. "Figure" to be "Fig.", subscript, etc )
3. Prepare figures in high-resolution
4. Explain all symbols in Figures such as red arrows (Figs. 2, 5, and 8), "###" (Figs. 7 and 8), and "SN" (Figs. 6 and 7)
5. Use abbreviation you defined in manuscript (lines 169 and 170) such as 1287-5p mimics instead miR-1287-5p mimics.
6. Please describe more detailed information on Materials and Methods to be able reproduce by others. (e.g.: brand and origin of all equipments, kits, reagents, software, secondary antibody, siRNA sequence information )

Experimental design

In Figs. 2, 5, and 8, please describe how you evaluated the neutrophils adhesion to HUVECs. It looks difficult to tell the neutrophils adhesion effects on exosome, mir-1287-5p, and siRNA from the light microscopic images.

In Fig. 6A, is there other transfection time other than 24 hrs? You mentioned "partly due to the limited transfection time" (line 301) therefore if you could prolong the transfection time, the clear cut data would be obtained?

Fig. 6B, it looks almost no-signals in lane of 1287-5p mimic of CBL picture whereas there is about 40% intensity of MNC in 1287-5p mimic. How did you measure the intensity of CBL?

In Figs. 6B, 7A, and 7C, regarding Western Blotting of CBL intensity compare to its control. If three data was compared, the amount of CBL looks like significantly different each other when normalized with b-tublin. The levels of CBL in HUVECs could be fluctuated by any reason?

Do you have any data that miRNA from green dots such as miR-129-1-3p (Fig. 3A) was transfected into HUVECs? The inhibition effects on inflammation and/or neutrophils adhesion are expected.

Validity of the findings

If you could addressed above points, entire manuscript validity will be better.

Additional comments

NA

Reviewer 2 ·

Basic reporting

Review: major revision recommended
Basic reporting
Language needs improvement, so that the international audience can clearly understand your text, please look closely at detailed comments /examples belo. Language is often unclear, unprecise, there are abundant grammar and spelling mistakes. The passive sentence structure makes it difficult to read and understand. I suggest you search help from a fluent English speaker or a professional language editing service
You should not use a plural “s” with miR-mimic or -inhibitor, if they used only one of each. Also abbreviations should not have a plural “s” like ANCA, HUVEC, miRNA etc
Basic conventions are neglected: you refer to CBL and it is unclear if they refer to the gene, the mRNA or protein. Please correct all paragraphs using unambiguous language: CBL-mRNA or CBL protein
You use different abbreviations for the same thing: HC, NC for control group
Intro & background: abstract is too short, there is room for more details, relevant background, precision in question and methods lacking. How many miRNAs were screened?
->For more details go to comment on abstract
The introduction should report on
- overview of miRNA regulation in ANCA-assoc. diseases, vasculitis or endothelial inflammation (missing)
- What is known of miR-1287-5p? in particular in context of vasculitis, endothelium or inflammation
1. This should be picked up in the discussion. 2. why was the focus on CBL (in silico target prediction results were not reported in full detail), what is the known pathophysiology of CBL and how do results of this study add to CBL mechanism in inflammation, vasculitis or endothelial inflammation?
Structure as per Peer J standard:
You should not repeat review of literature in the description of methods or results- this is part of the introduction and discussion

Standard sections: yes
Research question, relevance explained: yes
Self contained submission: in general : Yes
Original primary research: yes
Rigourous investigations to a high standard: see detailed comments below
Sufficient and detailed description of methods to replicate: no, needs improvement
Cite previous reviews: no: better selection of reviews regarding miRNA and exosomes in vasculitis and endothelial inflammation/autoimmunity recommended, you cite less-relevant reviews from cancer, metabolic disease etc
Conclusions: too speculative: you claim that you clarified the effect on CBL on vascular inflammation but these effects were not investigated. Unresolved questions not identified, future directions should be explained better: how might CBL be a target in MPA treatment- pathophysiological relevance/background/literature not discussed.
Reference section: full title of journals and DOI not provided
Experimental design/ methods incompletely reported, please present results in the same order as methods, structure methods and results logically: hypothesis -> computational target prediction -> molecular function (dual luciferase assay)-> cell culture
Validity of findings:
Main findings are not properly reported: number and list of all analysed miRNAs missing, number of patients, clinical characteristics, demographics not provided, and controls not reported, statistics not corrected for multiple testing
You should report all results of target prediction and explain how they conclude that CBL is the main relevant target, - provide supplemental file with data!
Ethic vote: yes
Figures
quality: often insufficient,
well labelled: no, insufficient, sometimes illegible,
description insufficient
Please restructure your figures, they are presented not in the correct order: legends are separated from images, legend 1 on page one, table 1 on page 2, legend 2 on page 3, fig.1 in page 4, fig. 2 in page 6…. This is unacceptable. Also, abbreviations in fig. Legends don’t match abbreviations in the text. In legends new abbreviations should be re- introduced as NC, INC as the reader can’t remember uncommon abbreviations or some of the abbreviations were never introduced in the text

Raw data/supplemental files: not supplied. Thank your for supplying all raw data, all data retrieved such as all investigated miRNA (volcanoplot), full list, explain cut-off levels/ fold changes, please use appropriate tests with corrections for multiple testing, show all p -values
Please provided patient numbers and clinical characteristic so the reader can know patients fulfilled MPA ACR/EULAR criteria. Please show basic demography of controls as well – were controls matched?

comments in detail
Fig.1A description lacking: what does the reader see, poor sharpness, cells are unrecognizable, what is the black „blob“ scale bar illegible (font size too small)
Fig 1B provide full length western blot with protein ladder as control (supplemental fig.)
Fig 1 c show both exosome size distribution for HC and MPA
Fi1d contains three images with different colours what does the reader see?, what are the red arrows?
Table 1: sequences of 1287 mimic and inhibitor and sequences of siRNAs missing
Sequences of CBL-3’UTR wildtype and mutation missing
Show results of targetScan, computational target prediction of binding strenghts missing
Fig. 2
legend is unprecise and incomplete: I suggest „(A) relative levels of proinflammatory factors and adhesion molecules IL-6, ….. and E-selectin in treated HUVECs ….
The description of the graph itself is missing: black bars show …. Grey columns indicate ……

Fig 2 B: descriptions missing what are the HUVECs, what are the neutrophils, what point the red arrows at? Neutrophils are unrecognizable, provide prove of neutrophils, larger scale/resolution image or specific staining
Fig 3
what is NC -exo? Use same abbrev. As in text, repeat full text for abbrev. As this is very difficult and confusing for the reader. State how many patients and healthy donors were investigated n=?
font size of axis illegible
Fig 3 A red dots: specify which miRs are up/downregulated, provide names in the graph o ras table or in the text. Log change not above 2, please explain how you defined a cut-off for significantly changed miRNA, usually it is > 2
Fig. 4
Ll 2 4: spelling/grammar: In HUVECs treated with …. adhesion molecules including IL-6, IL-8, MCP-1, ICAM-1, and E-selectin levels were….
Please explain abbrev again MNC, INC is unclear
Please show baseline endogenous miR-1287 expression in HUVESs without before tranfection of mimic, inhibitor or adding exosomes.
Fig. 5 Please use descriptive title not interpretation such as “Effect of transfecting HUVECs with miR-1287-5p mimic and inhibitor on neutrophil adhesion”.
Again, figure description is incomplete: what are the red arrows pointing at? How can we be sure it’s neutrophils. Image resolution, magnification insufficient -> improve image quality or stain with neutrophil marker
Fig. 6 Please use descriptive title not interpretation such as “Effect of miR-1287-5p transfection on CBL-mRNA”. Grammar: “(B) The expression of CBL at the protein levels in HUVECs….
Fig 6A: indicate “no significance”
Explain abbreviations, like lNC in fig6A-B5
Fig 6 B the use of 1287-5p inhibitor increases CBL protein compared to negative controls, this indicates endogenous miR1287 expression in HUVECs- please show RTqPCR for miR1287-5p in HUVECs without transfection
Fig. 7: what is NC exo…so far it always was refered to HC -exo, please be precise!
7b-d: Also si-NC is not explained, no description in methods…
Fig7c looks manipulated! Please repeat and deliver a full resolution image and provide the protein ladder in order to estimate the molecular weight

Tab. 1: sequences for mimic-1287, 1287- inhibitor and siRNA missing- please indicate


Abstract : only 1800 characters, 250 words, there is room for more precision:
Language: abundant spelling and grammar mistakes, passive sentence structures make it difficult to read -> get support from native speaker
Examples grammar/spelling/language – abstract only
Line 37 Background
Line 44 […]differential miRNA expression in MPA-exo compared to[…]
Line 47-48 […]A co-culture system […]was established
Line 52-54 […]Up-regulating […] and down-regulating had the opposite effect (it’s not a phenomenon….)
Content: insufficient background: why are miRNAs relevant in AAV ? What is he function of CBL, how could it contribute to AAV pathogenesis, what is known about miRNA dysregulation in AAV
Unprecise methods/errors in basic exosomal markers for ex TST101 instead of correct TSG101
Unclear what exact assays/methods were used?
Grammar: Unprecise language, use active sentences, not passive sentence structures
Basic numbers missing: how many miRNA were screened,. How many patients and healthy donors were included n=?
Line 45 „relationship“-> what exact mechanism or hypothesis did you want to investigate? Experimental design ? It is unclear if they investigated the expression of cbl-mRNA or CBL protein or both, if you refer to mRNA the abbreviation should be itacalized please indicate your methods and purpose of methods : lines 47-48 “A co-culture system was established…”-> in order to investigate what?
It seems you have a fundamental misunderstanding of miRNA up/downregulation. The latter would refer to an increased miRNA gene expression, but these methods of up/downregulation of miRNA genes was not used. You used an external exosomal transfer of miRNA transfection of mimics in order to increase miRNA levels. This should be corrected throughout all paragraphs of this manuscript. miRNA were not up- or downregulated by a genetic alteration but external miRNA or siRNA were added (transfected)

Experimental design

Materials and Methods
Should be structured logically. It all starts with in silico target prediction- the description of this computational method is missing
Please state how many samples from how many patients and HC were used, n =?
Please show baseline endogenous miR-1287 expression in HUVECSs without before tranfection of mimic, inhibitor or adding exosomes.
Please clarify which exosomal markers were used- is TST101 in line 141 correct- shouldn’t it be TSG101- please explain
Please determine cellular origin of exosomes: are the MPA-exo released from platelets, neutrophils, B-cells, endothelial cells ?
L 168ff Transfection
Methods are incomplete: information on 1287-mimic , inhibitor and three CBL siRNAs are not provided. You declare that a company did the work, but what sequences were used?
Only endogenous miR can be up or downregulated by blocking or stimulating miRNA gene expression, the transfection means to add external miRNA into the cells, but this is not an up or downregulation- be more careful with specific language.
L 174f siRNA: please specify sequences, show in tab 1
L175 HUVECS don’t decrease CBL expression, but siRNA transfection does reduce CBL-mRNA transcription- sentence is unclear: what do you want to say? How did you /which HUVECs did you select for further studies?
219 Dual-luciferase reporter assay -> To brief: how did you mutate cbl-mRNA, description of method missing, reference to fig 6c missing

220 A dual-luciferase reporter assay was performed to verify whether CBL is the target mRNA of miR-
221 1287-5p. Briefly, HEK293T cells were plated in 96-well plates and transfected with 3′-UTR wild-type
222 (WT) or mutant (MUT) CBL-harboring plasmids in the presence of miR-1287-5p mimic or MNC in strict
223 accordance to the manufacturer’s specifications. At 48 hours after transfection, the cell lysates were
Silencing si RNA experiments: What is NC, there is no “normal Control” in this setting, the description of silencing siRNA experiments and controls is missing in methods

Validity of the findings

Results
Should be structured better and should have same order of presentation as the methods
line 52 „CBL was identified as target of mir-1287-5p“ This is not correct as miRNA don’t target protein but mRNA. It is unclear how this „identification“ happened. Report the results of the dual luciferase reporter assay .
Characterization of exosomes
Show differences of sizes HC-exo vs MPA-exo-> refine fig1c, description of fig is insufficient, see comment on figures
what cells do the exosomes originate from: is there a difference between MPA and HC?
248 PCR. As shown in Figure 2A, the relative expression levels of IL-6, IL-8, and MCP-1 and of the adhesion molecules ICAM-1 and E-selectin were significantly
249 increased in HUVECs treated with MPA-exo compared to HUVECs treated with HC-exo (p < 0.05).
250 effects of MPA-exo on the expression of the adhesion molecules ICAM-1 and E-selectin were also
251 analyzed by qRT–PCR. Incubation of HUVECs with MPA-exo resulted in upregulation of ICAM-1 and
252 E-selectin mRNA expression (p < 0.05).
Please report effect size of changes between HC and MPA-exo: the fold changes were ……
The description of results for fig 2 C neutrophil adhesion are incomplete: how man high power fields were analysed. How much was the neutrophil adhesion increased ?
Language: 257 activation (better: transfer or treatment) could significantly increase endothelial inflammation (better proinflammatory cytokine release in HUVECs) and neutrophil adhesion to HUVECs.
259 The miRNA expression profile of plasma exosomes in MPA patients
Here you repeat a background but you fail to describe your results in detail: how many miRNAs were found, which specific miRNAs were significantly (corrected for multiple testing?) changed at what level. How did you define “significant change”. Please state which specific miRNAs were downregulated and which were upregulated at what fold change? -> Please discuss why a fold change below 2x is considered relevant, fig 3A semas to show log changes less than twofold? But legend is illegible, please use larger font size. Usually a cut-off >2fold change is considered physiologically relevant.
260 After being delivered into recipient cells, exosomal miRNAs participate in a diverse range of cellular
261 functions by regulating target genes. To characterize whether the miRNAs in MPA-derived exosomes
262 play an important role in the development of MPA, we used RNA sequencing to explore the miRNA
263 spectrum between healthy controls and MPA patients. Upon comparing global miRNA expression
264 profiles in MPA with those in healthy controls, we identified that 6 miRNAs were significantly
265 upregulated and 3 miRNAs were significantly downregulated in MPA. The volcano map displays the
266 overall distribution of differentially expressed miRNAs (Figure 3A). Moreover, we selected three
267 differentially expressed exosomal miRNAs of associated with MPA (miR-1303, miR-129-1-3p, and miR
268 1287-5p) for qRT–PCR to validate the sequencing data. Their expression levels were normalized to that
269 of cel-miR-39, as control. Compared with those in healthy controls, the expression levels of

273 MiR-1287-5p enhanced HUVECs inflammatory response and neutrophil adhesion
Here again you don’t differentiate between investigating mRNA or protein…their language always refers to protein, but methods were used for mRNA (qRT-PCR)
274 For further study, we selected miR-1287-5p, which was relatively abundant and showed
275 significantly differences in both healthy controls and MPA patients. We tested the changes in miR-1287
276 5p expression after the uptake of exosomes by HUVECs after 24 hours and found that the
277 levels of miR-1287-5p increased in the HUVECs (p < 0.05, Figure 4A). Subsequently, the effects of miR-
278 1287-5p on HUVEC’s inflammatory response and neutrophil adhesion were determined. The qRT–PCR results
279 shown in Figure 4B suggested that miR-1287-5p mimics predominantly increased the mRNA expression of IL-6,
280 IL-8 and MCP-1, and miR-1287-5p inhibitors significantly inhibited the mRNA expression of these factors
281 compared to that in the control group (p < 0.05), except there was marginal evidence for MCP-1 (p =
282 0.081). The mRNA expression of adhesion molecules was also investigated by qRT–PCR. In comparison with the
283 expression in the control group, ICAM-1 and E-selectin mRNA levels expression were markedly elevated after
284 treatment with miR-1287-5p mimics but were reduced when the HUVCEs were treated with the miR-1287-5p
285 inhibitor (Figure 4B, p < 0.05). The neutrophil adhesion assay results shown in Figure 5A&B revealed
286 that miR-1287-5p mimics promoted the adhesion of primed neutrophils to the HUVECsendothelium, which was
287 suppressed by the miR-1287-5p inhibitor (Figure 5C&D, p < 0.05).

289 CBL is a target gene of miR-1287-5p
Should be structured better and should have same order of presentation as the methods
Start with results of computational target prediction, mentioned all main results, not only cbl-mRNA, please explain how you chose CBL to be your candidate of interest, you should use different in silico target prediction algorhitms to verify the targetScan results. What were the differences between different algorithms?
Suggested order of methods and results
a) In silico target prediction
b) Functional assays
a. On molecular level: Luciferase reporter assay
b. On cellular level CBL mRNA in cultured cells
c. CBL protein in cultured cells
You should not repeat review of literature in the description of results- this is part of the introduction and discussion
291 inducing HUVEC’s inflammatory response and neutrophil adhesion. A reviewing of literature suggested that miR-
292 1287-5p mimics may indirectly upregulate the expression of EphA2 in HUVECs, which is probably
293 achieved by inhibiting CBL (Satake et al., 2021). In addition, the presence of A binding
294 site in CBL-mRNA for miR-1287-5p was predicted using TargetScan website
295 (http://www.targetscan.org/vert_72/). Therefore, we speculated that miR-1287-5p may play a role in
296 HUVEC’s inflammatory response and neutrophil adhesion by targeting CBL. To verify our hypothesis, we
297 determined the CBL-mRNA levels of CBL using qRT–PCR after HUVECs were transfected with
298 miR-1287-5p mimics or a miR-1287-5p inhibitor for 24 hours. As shown in Figure 6A, CBL was
299 downregulated in HUVECs overexpressing miR-1287-5p (p < 0.05). The CBL-mRNA level in HUVECs
300 increased after transfection with the miR-1287-5p inhibitor, but the difference did not reach statistical
301 significance (p = 0.058), partly due to the limited transfection time. Western blot analysis also
302 demonstrated that HUVEC treated with miR-1287-5p mimics for 48 hours exhibited a marked reduction
303 in CBL protein expression, while CBL protein was upregulated after treatment with miR-1287-
304 5p inhibitors (Figure 6B). Eventually, the results of the dual luciferase reporter assay suggested that
305 cotransfection with miR-1287-5p mimics decreased the luciferase activity of the wild-type
306 3′-UTR of CBL (CBL-Wt, p < 0.001), whereas this effect was reversed when the luciferase activity of the
307 CBL 3′-UTR mutant type (CBL-Mut) was not inhibited detected (p = 0.0031, Figure 6C&D). In brief, these results
308 indicate that CBL -3’UTR-mRNA contains a binding site targeted by miR-1287-5p.
309

310 Downregulation of CBL promoted inflammation and neutrophils adhesion of HUVEC

311 To determine whether MPA-exo affected the protein expression of CBL in HUVECs, we cultured
312 HUVECs in the presence of HC-exo or MPA-exo for 48 hours. As shown in Figure 7A, treatment with
313 MPA-exo decreased the expression of CBL in HUVECs (p < 0.05). Subsequently, in order to explore the
314 effect of CBL in inflammation and neutrophil adhesion on HUVECs, siRNAs (si-001, si-002, si-003)
315 were used to silence CBL mRNA expression. Upon Combining qRT–PCR and western blot results, we
316 found that the most efficient siRNA for CBL silencing was si-003 (Figure 7B&C). HUVECs transfected
317 with si-CBL (si-003) exhibited significantly increased mRNA expression of IL-6, IL-8, MCP-1, ICAM-1 and E
318 selectin after 48 hours compared with the NC group (p < 0.05, Figure 7D). Blocking CBL expression also
319 contributed to neutrophil adhesion on HUVECs (p < 0.05, Figure 8A&B). In addition, these dysfunctions
320 induced by si-CBL were reversed by the miR-1287-5p inhibitor (p < 0.05). Taken together, the results
321 revealed that CBL inhibition was associated with an increased inflammatory response and increased neutrophil
322 adhesion, which demonstrated the similar effect of the miR-1287-5p mimics on HUVEC dysfunction.

Discussion-This needs to be discussed:

methods
-What impact has your chosen method of exosome purification on its cargo/miRNA
Comment on hypothesized role of CBL in inflammation, endothelitis, vasculitis- how would you interprete your results in the pathophysiological context of AAV, how could you prove your hypothesis, how could it help to develop new AAV therapies- please specifiy. Your current remarks are unclear and to unspecific. See comment above: As you don’t know the pathophysiology of CBL you can’t say that this is the “cause”- too much interpretation! -> discuss, what could be the proinflammatory mechanisms and suggest functional experiments for the future outlook

-Fig 6 B the use of 1287-5p inhibitor increases CBL protein compared to negative controls, this indicates endogenous miR1287 expression in HUVECs- please show RTqPCR for miR1287-5p in HUVECs without transfection
Please discuss putative mechanisms and report the literature on miR1287-5p and CBL in inflammation, endthelitis/vasculitis,

Please check literature for miRNA profiles in MPA/AAV- how do you explain differences in literature and the miRNA profile in this study; again: report/name all miRNA you analysed (volcanoplot shows hundreds of miRNAs, provide supplemental data on what you did) and compare to literature
Please indicate if you found mir-30c, mir-135a, and mir-27a, miR- 1323 and miR-155 and others to be changed in MPA as those were shown to regulate CBL as well
Please comment on hypoxia mediated regulation of CBL and hypoxia in vasculitis
You need to explain why is CBL protein expression is higher after 1287 inhibitor treatment compared to the miR-negative control MNC fig 6 B-> they should check HUVEC for endogenous miR1287 expression

325 The pathophysiological mechanisms associated with MPA are complex and are not entirely
326 understood. An increasing number of studies have confirmed an inflammatory damage to
327 the vascular endothelium caused by leukocyte infiltration,
328 a pathological hallmark of MPA (Nagao et al. 2011; Rymarz et al. 2021). In the present
329 study, we hypothesized a novel mechanism through which the exosomal miR-1287-5p mediates
330 vascular endothelial inflammatory damage via inhibition of CBL, which may provide a novel molecular target for the treatment

333 certain cargo (such as miRNAs) into recipient cells, which has attracted extensive attention for scientific
334 research in recent years (Guay & Regazzi 2017). Previous studies have indicated that exosomal miRNAs
335 participate in cell apoptosis, metabolism, inflammation, and proliferation (Li et al. 2017; Martinez-Bravo
336 et al. 2017). Further investigations have shown that exosomal miRNAs are involved in the progression of
339 triggering injury and inflammation (Zhang et al. 2021). MiR-1-3p in sepsis plasma-derived exosomes
340 plays a critical role in regulating endothelial inflammation by targeting SERP1 (Gao et al. 2021). In

353 MPA. The results showed that MPA-derived exosomes induced an inflammatory response in HUVEC with neutrophil
354 adhesion. Furthermore, we identified by RNAseq and qRT-PCR that miR-1287-5p was significantly upregulated in MPA plasma
355 derived exosomes compared exosomes from HC. We also verified an increase in the exosomal miR-1287-5p level in the plasma of MPA
356 patients by qRT-PCR. In addition, we observed the transfer of MPA-derived exosomes into the cytoplasm
357 of HUVECs, and we showed a significant uptake of exosomal miR-1287-5p level by HUVECs treated with
358 MPA-derived exosomes. The results might indirectly indicate that plasma exosomes in MPA may serve
359 as carriers for the transfer pro-inflammatory miRNAs such as miR-1287-5p to the site of vasculitis. It was reported that exosomes play a critical role
360 in the development of autoimmune vasculitis (Wu et al. 2019). Circulating exosome levels were correlated with disease activity in systemic lupus erythematosus (Lee et al. 2016).
362 Therefore, we hypothesize that circulating exosomes might be absorbed by endothelial cells in MPA and
363 participate in the vasculitis progression through a proinflammatory effect of miR-1287-5p.
364 As a member of the miRNA family, miR-1287-5p was first identified in tumors. It has been revealed
365 that miR-1287-5p can act as a significant suppressor in different cancers, including breast cancer
366 (Schwarzenbacher et al. 2019), Cervical Cancer (Ji et al. 2020), pancreatic cancer (Zhang et al. 2020), and
367 non-small cell lung cancer (Shanshan et al. 2021). MiR-1287-5p regulates cell growth, apoptosis,
368 ferroptosis, and invasion by targeting phosphoinositide 3-kinase CB, glutathione peroxidase 4, and Y-box
369 binding protein 1 (Cui et al. 2020; Schwarzenbacher et al. 2019; Shanshan et al. 2021). Recent studies
370 have proposed that miR-1287-5p also participates in other biological functions: By overexpressing transfecting miR
371 1287-5p – mimic increases caspase-3 activity was increased and angiogenesis was inhibited in HUVECs
372 targeting angiopoietin-1 (Sanchez et al. 2019). According to Zhang et al., miR-1287-5p is associated with
373 HBV-induced immune suppression (Zhang & Wang 2020). Satake et al. demonstrated that circulating
374 miR-1287-5p plays an important role in the risk of end-stage kidney disease in patients with diabetes. In
380 on vascular inflammation. The results of our study may support with this hypothesis. Elevated
387 To comprehensively In order to hypothesise a molecular mechanisms by which miR-1287-5p influences the
388 HUVEC inflammatory response, we predicted possible target genes of miR-1287-5p using the TargetScan
389 algorithm. MiRNA induce posttranscriptional gene silencing by binding to the 3’UTR terminus of target
390 genes, mostly interrupting mRNA expression and/or inhibiting protein synthesis. Bioinformatic prediction suggested
391 CBL to be a target of miR-1287-5p . The dual-luciferase
392 reporter assay supports this hypothesis as miR1287-5p reduced the activity of the luciferase / CBL-3’UTR construct . In addition, both western blotting and qRT–PCR indicate that miR
393 1287-5p mimics downregulated the expression of CBL protein in HUVECs, while a miR-1287-5p inhibitor
394 enhanced the expression of CBL protein. CBL is a 120 kDa protein that was identified as a negative regulator of
395 receptor tyrosine kinase signalling through ubiquitylation and natural killer cell function (Wang et al.
396 2002). It has been reported that CBL might alleviate endothelial dysfunction in patients with diabetes
397 mellitus by inactivating the JAK2/STAT4 signaling pathway (Jin et al. 2021). A study by Duan et al.
398 demonstrated that CBL deficiency in dendritic cells aggravated intestinal inflammation by inhibiting
399 fungus-induced noncanonical NF-κB activation (Duan et al. 2021). Chen et al. indicated that miR-216a
400 could promote odontoblast differentiation by targeting CBL (Chen et al. 2020). In this study, we was found
401 that MPA-derived exosomes decreased the expression of CBL protein, and the reduction in CBL protein

Conclusion:
comment on CBL physiological function and what impact it might have in AAV
the conclusion must not be that blocking CBL increased cyotkines or neutrophil adhesion- these experiments were not done…there might be other mechanisms – you should be aware that an association is not a sufficient cause

407The current study showed that miR-1287-5p was significantly enriched in MPA exosomes.
408 In a HUVEC cell model , it mediated the expression of inflammatory factors and adhesion molecules, and promoted neutrophil
409 adhesion to HUVECs through its target gene CBL, associated with a vascular inflammatory environment . Thus , miR1287-5p might
410 contribute to the acute injury in MPA. Our study demonstrated for the first time, that
411 miR-1287-5p targets CBL. and may provide potential new targets for treating MPA-induced
412 vascular inflammatory injury.

Additional comments

for all comments/complete review please check uploaded pdf

Further remarks on language, grammar and spelling
36 Abstract
37 Background: Microscopic polyangiitis (MPA) is characterized by inflammatory necrosis of small sized
38 vessels. Inflammatory environment around the vessel wall caused by leukocytes infiltrations is one of the
39 characteristic histopathological features of MPA; however, the pathogenic mechanisms are not fully
40 understood. Herein, we identified the effect of exosomal miRNAs on vascular endothelial cells.
41 Method: Plasma exosomes from patients with MPA (MPA-exo) and healthy controls (HC-exo) were
42
isolated. The effects of vascular endothelial cells were evaluated by examining levels of IL-6, IL-8, MCP-
43 1, ICAM-1 and E-selection and by neutrophil adhesion assay. The Exosome small RNA sequencing was
44 conducted to screen differential miRNA expression in MPA-exo and HC-exo. Dual luciferase reporter gene
45 assays were performed to confirm the relationship between miR-1287-5p and Casitas B-lineage
46 Lymphoma (CBL-mRNA). The expression of CBL was knockdown by siRNA to explore the underlying
47 mechanisms. A co-culture system of plasma-derived exosome and human umbilical vein endothelial cells
48 (HUVECs) were established.
49 Result: Our results illustrated that MPA-exo significantly increased the expression of inflammatory
50 factors, adhesion molecules and CBL protein in HUVECs and promoted the adhesion of neutrophils to
51 HUVECs. A high abundance of miR-1287-5p was observed in the exosome obtained from MPA plasma compared to HC-exo.
52 CBL was identified to be a target of miR-1287-5p. Up-regulated miR-1287-5p or silencing CBL
53 promoted inflammatory injury and neutrophil adhesion, and down-regulated miR-1287-5p reversed this
54 phenomenon.
55 Conclusion: Our study revealed that MPA-derived exosomes were involved in the intercellular transfer
56 of miR-1287-5p and subsequently promotion the development of acute injury in MPA. MiR-1287-5p and
57 CBL may be promising therapeutic approach for MPA-induced vascular inflammatory injury.
58
60 Introduction
61Anti-neutrophil cytoplasmic autoantibody (ANCA)-associated vasculitides (AAV) are a group of
62 disorders characterized by inflammation and destruction of small-sized vessels leading to endothelial
63 injury and tissue damage, and accompanied by the presence of ANCAs in serum. It is well established
64 that myeloperoxidase (MPO) and proteinase 3 (PR3) are two major target antigens of ANCAs (Nakazawa
65 et al. 2019). AAV is divided into three clinical phenotypes: granulomatosis with polyangiitis (GPA),
66 microscopic polyangiitis (MPA), and eosinophilic granulomatosis with polyangiitis (EGPA) (Jennette &
67 Nachman 2017). AAV is an uncommon disease with notable ethnity-based differences, where
68 MPO- ANCA positive and MPA predominates in East Asian countries, including China (Geetha & Jefferson 2020).
69 Massive inflammatory injury to vascular endothelial cells can result in necrotizing vasculitis, which is one
70 of the defining histopathological features of MPA. Neutrophils are recognized as effector cells
71 responsible for endothelial damage in the acute AAV-related injury (Al-Hussain et al. 2017). Neutrophils
….
95 This study to investigated the characteristics of miRNA expression profiles in
96 circulating exosomes from MPA patients. Exosomes were isolated from plasma for small-RNA
97 sequencing analysis, and the candidate miRNAs responsible for MPA were identified by comparing the
98 miRNA expression differences between MPA patients and healthy controls. In this study, miR-1287-5p
99 was highly expressed in MPA-derived exosomes derived from active MPA in the active stage. Considering the important role of
100 inflammatory injury of endothelial cells in the pathogenesis of MPA, we speculated that miR-1287-5p from
101 carried by circulating exosomes from active in the active stage of MPA may be absorbed by vascular endothelial cells
102 and contribute to inducing endothelial cell inflammation and neutrophil adhesion.
110 … Patients with serious infections
117 The exosomes were purified from plasma from MPA patients or HCs as previously described with
118 minor changes (Chen Zhang1*, 2020). Briefly, Peripheral blood samples were collected in anticoagulant
119 tubes containing EDTA and centrifuged at 3,000×g for 15 min at 4 ℃ to obtain plasma. The plasma was
120 then centrifuged at 12,000×g for 45min, followed by filtration through a 0.22- μm filter. The collected
121 supernatant was subjected to ultracentrifugation at 120,000×g for 70min at 4 ℃. The exosome pellet was
227 ….The data are ….

Annotated reviews are not available for download in order to protect the identity of reviewers who chose to remain anonymous.

---

## Round 0.2 · Minor Revisions

Please address the comments from Reviewer 2 before the final publication, and we will accept your manuscript.

Reviewer 1 ·

Basic reporting

Authors have corrected their manuscript by responding to all reviewer's comments properly.

Experimental design

In the revised Methods section, it became easier to understand how to carry out experiments and obtain data.

Validity of the findings

In the discussion and conclusion, appropriated citations and data reflection are included and entire flow is much easier to read than original manuscript.

Reviewer 2 ·

Basic reporting

line 216: please correct table reference, primer seq. are in tab 2 not 1

line 314: spelling mistake neutrophil/HUVCE, please correct

figure 4 (B) caption: "...the mRNA expression of miR-1287-5p..."
comment: delete mRNA as this is not the same as miRNA. correct is: "... the expression of miR-1287-5p...."

figure 7 caption, line 5: please correct spelling mistake

Experimental design

Material and methods ll 119-120
regarding patients: I don't understand why you excluded only PR3-ANCA positive patients who had infections. It is important to exclude MPO-ANCA patient with infections, as infection can change the inflammatory response and miRNA expression profiles. Please correct : I guess you wanted to say " we excluded patients with serious infections and patients who were positive for PR3-ANCA..."

Validity of the findings

Table 1: hemoglobin levels are incorrect: normal values are 12-15 g/dL, 73 g/dl is impossible. wrong unit?


please add to discussion that you only measured cytokine mRNA and not protein that promotes the inflammatory environment, you can assume that mRNA would be translated to protein but you did not prove this.

discussion / table one : please comment on the treatment your patients had. Please discuss that for ex. steroid very quickly change miRNA expression . Also minor technical differences or sample storage procedures might change results. These are far better explanation why results are not reproducible compared to Wang et.al. I do not accept your statement that it was due to different ethnicity because that was a comparable chines cohort.

ll 401/402: "... A previous study..." please correct: this is not a single study but a review article that describes the impact of EV in vasculitis

Additional comments

Dear authors,
congratulations to a big improvement after re-editing and correcting language and supplyind more detailed information. It is much easier to read and to understand. There are only few minor errors/questions left and you just need to add few comments to the discussion.
Thank you for your effort!

---

## Round 0.3 · accepted · Accept

All of the reviewers' comments have been addressed.